# Spatial and temporal organization of RecA in the *Escherichia coli* DNA-damage response

Harshad Ghodke[1,2], Bishnu P Paudel[1,2], Jacob S Lewis[1,2], Slobodan Jergic[1,2], Kamya Gopal[3], Zachary J Romero[3], Elizabeth A Wood[3], Roger Woodgate[4], Michael M Cox[3], Antoine M van Oijen[2]*

[1]Molecular Horizons and School of Chemistry and Molecular Bioscience, University of Wollongong, Wollongong, Australia; [2]Illawarra Health and Medical Research Institute, Wollongong, Australia; [3]Department of Biochemistry, University of Wisconsin-Madison, Madison, United States; [4]Laboratory of Genomic Integrity, National Institute of Child Health and Human Development, National Institutes of Health, Bethesda, United States

**Abstract** The RecA protein orchestrates the cellular response to DNA damage via its multiple roles in the bacterial SOS response. Lack of tools that provide unambiguous access to the various RecA states within the cell have prevented understanding of the spatial and temporal changes in RecA structure/function that underlie control of the damage response. Here, we develop a monomeric C-terminal fragment of the λ repressor as a novel fluorescent probe that specifically interacts with RecA filaments on single-stranded DNA (RecA*). Single-molecule imaging techniques in live cells demonstrate that RecA is largely sequestered in storage structures during normal metabolism. Upon DNA damage, the storage structures dissolve and the cytosolic pool of RecA rapidly nucleates to form early SOS-signaling complexes, maturing into DNA-bound RecA bundles at later time points. Both before and after SOS induction, RecA* largely appears at locations distal from replisomes. Upon completion of repair, RecA storage structures reform.
DOI: https://doi.org/10.7554/eLife.42761.001

**Competing interests:** The authors declare that no competing interests exist.

## Introduction

All cells possess an intricately regulated response to DNA damage. Bacteria have evolved an extensive regulatory network called the SOS response to control the synthesis of factors that protect and repair the genome. Processes coordinately regulated within the SOS response include error-free DNA repair, error-prone lesion bypass, cell division, and recombination.

The RecA protein is the master regulator of SOS, with at least three distinct roles. First, RecA forms a ternary complex with single-stranded DNA (ssDNA) and ATP to form the activated RecA*. RecA* catalyzes auto-proteolysis of the transcriptional repressor LexA to induce expression of more than 40 SOS genes (*Courcelle et al., 2001*; *Fernández De Henestrosa et al., 2000*; *Kenyon and Walker, 1980*; *Little and Mount, 1982*; *Little et al., 1981*). RecA* thus uses the ssDNA generated when replication forks encounter DNA lesions as an induction signal (*Sassanfar and Roberts, 1990*). Second, along with several other accessory proteins, RecA mediates error-free recombinational DNA repair at sites of single-strand gaps, double-strand breaks (DSBs) and failed replisomes (*Cox et al., 2000*; *Kowalczykowski, 2000*; *Kuzminov, 1995*; *Lusetti and Cox, 2002*). Third, the formation and activity of active DNA Polymerase V complex capable of lesion bypass requires RecA* (*Jaszczur et al., 2016*; *Jiang et al., 2009*; *Robinson et al., 2015*).

RecA is a prototypical member of a class of proteins that are critical for genomic stability across all domains of life (*Baumann et al., 1996*; *Bianco et al., 1998*; *Lusetti et al., 2003b*; *San Filippo et al., 2008*; *Sung, 1994*). In higher organisms, including humans, the homologous protein Rad51 supports error-free double-strand break repair by catalyzing strand exchange much like the RecA protein does in eubacteria (*Baumann et al., 1996*; *Sung, 1994*). Mutations in human Rad51 and accessory proteins have been implicated in carcinomas and Fanconi anemia (*Chen et al., 2015*; *Kato et al., 2000*; *Prakash et al., 2015*). Unsurprisingly, RecA and related recombinases are highly regulated, with a variety of accessory proteins governing every facet of their multiple functions (*Cox, 2007*). Directed-evolution approaches can be used to enhance the catalytic activities of recombinases in cells (*Kim et al., 2015*). However, RecA functional enhancement has a cost, disrupting an evolved balance between the various processes of DNA metabolism that share a common genomic DNA substrate (*Kim et al., 2015*). Many deleterious genomic events occur at the interfaces between replication, repair, recombination, and transcription.

An understanding of how organisms maintain genetic integrity requires an examination of the protein actors in their native cellular environments. In response to DNA damage, transcription of the *recA* gene is upregulated ten-fold within minutes (*Courcelle et al., 2001*; *Renzette et al., 2005*). Using immunostaining, the copy number of RecA in undamaged cells has been estimated to be about 7000–15,000 per cell, increasing to 100,000 per cell upon triggering the DNA-damage response (*Boudsocq et al., 1997*; *Stohl et al., 2003*). Visualization of C-terminal GFP fusions of wild-type and mutant *recA* alleles placed under the native *recA* promoter in *E. coli* have revealed that RecA forms foci in cells (*Lesterlin et al., 2014*; *Renzette et al., 2005*; *Renzette et al., 2007*). Interpretation of the localizations observed in these experiments has been clouded by three issues: (1) RecA fusions to fluorescent proteins have consistently resulted in proteins with reduced function (*Handa et al., 2009*; *Renzette et al., 2005*), making interpretation of the localizations revealed by these tagged proteins highly challenging. (2) This issue is further complicated by the fact that fluorescent proteins do not behave as inert tags and can influence intracellular localization in bacterial cells (*Ghodke et al., 2016*; *Ouzounov et al., 2016*). Indeed, *Bacillus subtilis* RecA tagged with GFP, YFP and mRFP yielded different localizations in response to DNA damage (*Kidane and Graumann, 2005*). These challenges do not come as a surprise since both N- and C-terminal ends are important for RecA function and localization (*Eggler et al., 2003*; *Lusetti et al., 2003b*; *Lusetti et al., 2003a*; *Rajendram et al., 2015*). (3) At least *in vitro*, untagged RecA has a remarkable ability to self-assemble, into different complexes that form on single-stranded DNA (RecA*), on double-stranded DNA, or are free of DNA (*Brenner et al., 1988*; *Egelman and Stasiak, 1986*, *Egelman and Stasiak, 1988*; *Stasiak and Egelman, 1986*). The properties of these assemblies are often determined by the state of hydrolysis of associated ATP. Thus, unambiguous assignment of the molecular composition of RecA features in live cells has been difficult.

In the absence of DNA, RecA can polymerize to form aggregates of various stoichiometry to yield dimers, tetramers, 'rods' and 'bundles' (*Brenner et al., 1988*). Some of these states may have a physiological relevance: RecA fusions with the best functionality have revealed DNA-free aggregates that are confined to the cellular poles, outside of the nucleoid and associated with anionic phospholipids in the inner membrane (*Rajendram et al., 2015*; *Renzette et al., 2005*). These DNA-free aggregates were hypothesized to be 'storage structures' of RecA, although their functionality in and relevance to the DNA damage response remain unclear.

Early electron-microscopy (EM) studies revealed that multiple dsDNA-RecA-ATPγS filaments could also associate to form structures confusingly termed as 'bundles' (*Egelman and Stasiak, 1988*). This study also identified that ssDNA-RecA-ATPγS filaments could aggregate together. Electron microscopy of cells revealed that RecA appeared to form 'bundles' that were aligned next to the inner membrane in cells after DNA damage (*Levin-Zaidman et al., 2000*). In cells carrying an additional allele of wild-type RecA at a secondary chromosomal locus to increase overall RecA function, long RecA structures called 'bundles' were formed during double-strand break repair (*Lesterlin et al., 2014*). These bundles are similar to RecA structures called 'threads', that nucleate at engineered double-strand breaks in *Bacillus subtilis* (*Kidane and Graumann, 2005*). RecA bundles form after SOS induction by other means than double-strand breaks, and also then interact with anionic phospholipids in the inner membrane (*Garvey et al., 1985*; *Rajendram et al., 2015*). The appearance of elongated RecA* foci after treatment with ultraviolet (UV) radiation has not always been associated with bundle formation (*Renzette et al., 2007*). It should be noted that whereas

assemblies of RecA observed *in vivo* have been variously referred to as filaments, threads or bundles, their correspondence to the *in vitro* observations of RecA aggregates referred to as 'rods' or 'bundles' remains unclear.

Due to the similar morphology of the fluorescence signal arising from these various DNA-bound repair or DNA-free storage structures, teasing out dynamics of individual repair complexes in live cells has proven difficult. The limited functionality of RecA fusion proteins utilized to date also raises concerns about the relationship of the observed structures to normal RecA function. Several fundamental questions remain unanswered: When and where does SOS signaling occur in cells? How is excess RecA stored?

In this work, we describe the development of a probe that specifically visualizes RecA structures on DNA, and utilize it as part of a broader effort to provide a detailed time line of RecA structural organization in living cells after DNA damage. With the objective of selectively localizing DNA-bound and ATP-activated RecA* as a key repair intermediate inside living cells, we produced a monomeric, catalytically dead N-terminal truncation of the bacteriophage λ repressor CI (mCI; *CI 101–229, A152T, P158T, K192A*) that retains the ability to bind RecA-ssDNA filaments. Removal of the N-terminal domain renders the mCI unable to bind DNA, leaving only RecA* as a binding partner. Using both untagged and fluorescently labeled mCI constructs, we document the effects of mCI *in vitro* and *in vivo*. We then use mCI as well as the most functional RecA-GFP fusion protein variants to distinguish between the various types of RecA structures and follow their behavior through time. In addition, we examine the location of RecA* foci formed in the nucleoid in relation to the location of the cellular replisomes. Our results reveal how the activity of RecA is regulated upon triggering of the SOS pathway and identify the various states of RecA that are relevant throughout the damage response.

## Results

### Experimental setup for live-cell imaging

Damage in the template strand can result in the stalling or decoupling of replication, leading to the accumulation of single-stranded DNA. This ssDNA provides a RecA nucleation site to form RecA*, the structure that amplifies the cellular signal for genetic instability (*Figure 1A*). In response to DNA damage, transcription of the more than 40 SOS-inducible genes is de-repressed upon cleavage of the LexA repressor including *recA* (*Courcelle et al., 2001*). Because production of RecA occurs rapidly after damage, it is critical to observe live cells at early time points with high temporal resolution after SOS induction.

With the objective of characterizing the spatial and temporal organization of RecA in cells during SOS induction, we performed time-lapse imaging of individual *E. coli* cells immobilized in flow cells using a variety of fluorescent probes (See Materials and methods for details of imaging). This setup enabled us to monitor growing cells with nutrient flow at 30°C, while keeping the cells in place to support long-term, time-lapse imaging of individual cells. A quartz window in the flow cell enabled us to provide in situ UV irradiation with a defined dose (20 $Jm^{-2}$) at the start of the experiment. Following this, we monitored fluorescence every 5 min over the course of 3 hr by wide-field acquisition (*Figure 1B*). For this study, we chose to induce SOS with UV for two key reasons: first, UV light is a strong inducer of the SOS response, and second, a pulse of UV light serves to synchronize the DNA damage response in cells that are continuously replicating DNA without the need for additional synchronization.

### Characterizing activity of the *recA* promoter during the SOS response

First, we set out to characterize the temporal activity of the SOS inducible *recA* promoter alone in wild-type MG1655 cells in response to UV radiation. We imaged cells that express fast-folding GFP from the *gfpmut2* gene placed under the *recA* promoter on a low-copy reporter plasmid, with a maturation time of less than 5 min ('pRecAp-gfp' cells; strain# HG260; supplemental table 2 in *Supplementary file 1*) (*Kalir et al., 2001*; *Zaslaver et al., 2006*). The copy number of this reporter plasmid has been shown to remain unchanged following ultraviolet radiation (*Ronen et al., 2002*). The cells retain the chromosomal copy of the wild-type *recA* gene. Measurements of the mean fluorescence intensity of cytosolic GFP in pRecAp-gfp cells exhibited a gradual increase peaking at

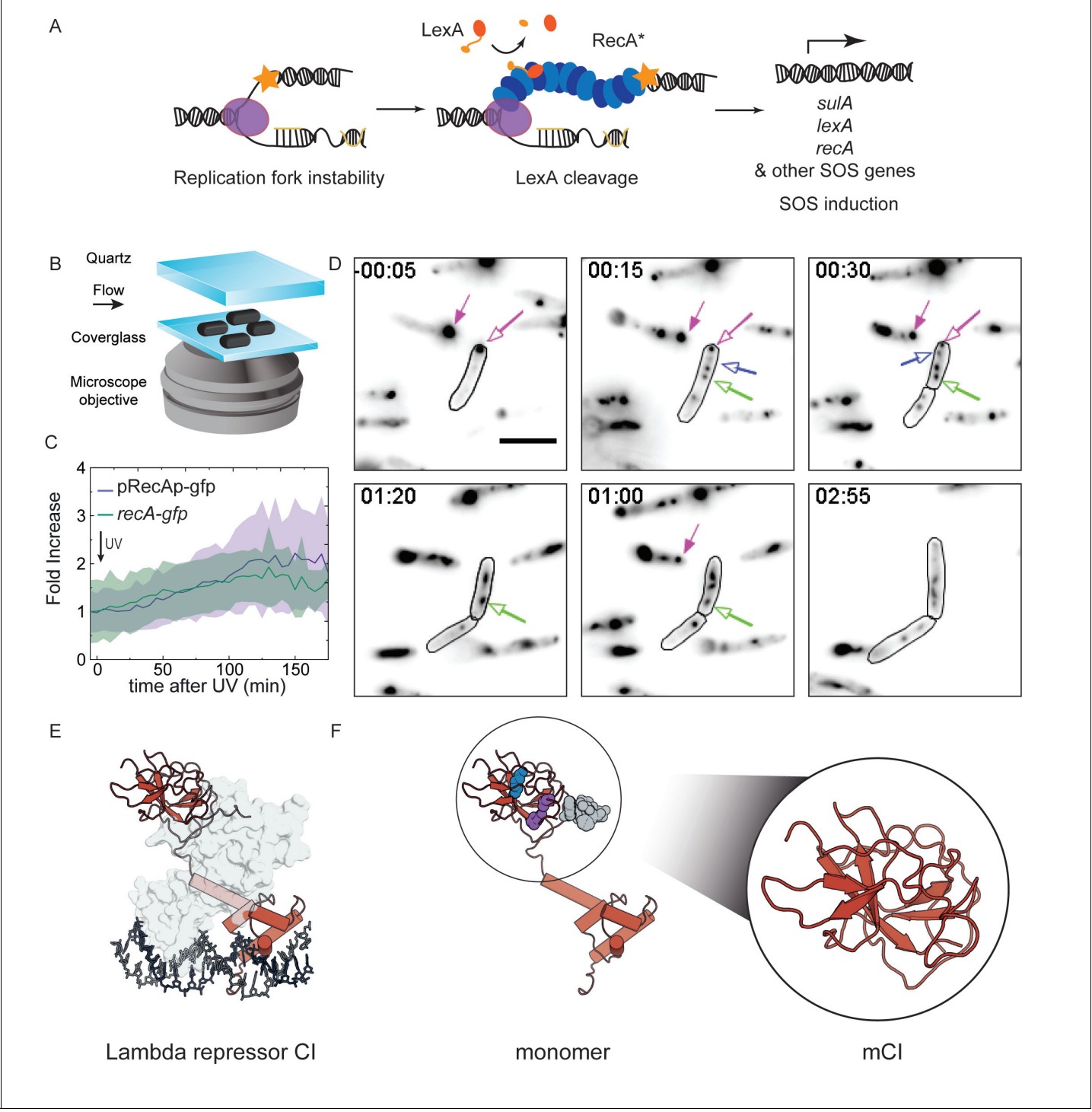

**Figure 1.** RecA forms different intracellular structures in response to UV irradiation. (A) Consensus model for SOS induction after DNA damage, illustrating the formation of ssDNA-containing RecA* filaments at sites of stalled replication forks. These RecA* filaments induce the SOS response by promoting cleavage of LexA. (B) Schematic of flow-cell setup for live-cell imaging. (C) Plots of relative increase in mean intensity of GFP in pRecAp-gfp *cells* (purple, strain# HG260) or RecA-GFP expressed from the native chromosomal locus (*recA-gfp* cells). Cells are irradiated with 20 Jm$^{-2}$ of UV at $t = 0$ min. Shaded error bars represent standard deviation of the mean cellular fluorescence measured in cells across the population. Between 50–200 cells were analyzed from 30 fields of view at each time point and two independent experiments were performed for each condition. See also *Figure 1— video 1*. (D) Imaging of *recA-gfp* cells (strain# HG195) reveals that RecA-GFP forms foci of various morphologies at different stages during the SOS response upon exposure to 20 Jm$^{-2}$ of UV. Magenta arrows indicate foci that are present before damage and disappear during the SOS response. Blue arrows indicate foci that appear after damage. Green arrow represents a focus that converts into a bundle. Cell outlines are provided as a guide to

*Figure 1 continued on next page*

*Figure 1 continued*

the eye. At least two independent experiments were performed with 30 fields of view at each time point. Stills from *Figure 1—video 2* are presented here. Scale bar corresponds to 5 μm. (E) Crystal structure of the operator bound dimeric λ repressor CI (PDB ID: 3BDN). (F) Monomer of CI showing the catalytic lysine (K192, purple), residues that mediate dimerization (A152 and P158, blue), and the C terminus involved in dimerization (grey). Inset shows the monomeric C-terminal fragment 'mCI' defined as CI(101–229, A152T P158A and K192A) used in this study.

DOI: https://doi.org/10.7554/eLife.42761.002

The following videos are available for figure 1:

**Figure 1—video 1.** Time-lapse acquisition of MG1655/pRecAp-GFP cells.

DOI: https://doi.org/10.7554/eLife.42761.003

**Figure 1—video 2.** Time-lapse acquisition of recA-gfp cells.

DOI: https://doi.org/10.7554/eLife.42761.004

approximately 135 min, with a maximum that corresponded to a two-fold increase compared to the initial fluorescence intensity (*Figure 1—video 1* and *Figure 1C*). After UV exposure, the accumulation of the fluorescent reporter protein reaches a maximum only after more than two hours. By extension, this gradual increase is used here to define the time period during which cellular RecA concentration is increasing after UV treatment.

Next, we imaged MG1655 cells that carry a *recA-gfp* fusion allele expressed from the *recAo1403* operator in place of the wild-type chromosomal copy of *recA* ('*recA-gfp*' cells; strain# HG195; supplemental table 2 in *Supplementary file 1*). The *recAo1403* promoter increases the basal (non-SOS) level of *recA* expression by a factor of 2–3 (*Rajendram et al., 2015*; *Renzette et al., 2005*). Despite the higher expression level, cells expressing this RecA-GFP fusion protein are deficient in RecA functions; notably, these cells exhibit a three-fold lower survival in response to UV irradiation, and ten-fold lower ability to perform recombination (*Renzette et al., 2005*). Additionally, these cells exhibit delayed kinetics of SOS induction but are still able to induce the SOS response to the same extent as wild-type cells (*Renzette et al., 2005*). In response to UV irradiation, GFP fluorescence in *recA-gfp* cells increased after DNA damage and peaked at approximately 130 min (*Figure 1—video 2* and *Figure 1C and D*). Thus, the kinetics of the observed increase in the levels of chromosomally expressed RecA-GFP fusion protein are the same as those of the increase seen with the plasmid-based *gfpmut2* reporter under control of the *recA* promoter.

Measurements of the abundance of the *recA* transcript after SOS induction have revealed a ten-fold increase within minutes after irradiation with UV in cells grown at 37°C upon exposure to 40 Jm$^{-2}$ of UV (*Courcelle et al., 2001*). In bulk experiments, the amount of RecA protein has been shown to attain a maximum at 90 min after introduction of damage (*Salles and Paoletti, 1983*). Our live cell experiments conducted at 30°C revealed lower fold increases in fluorescence than the 10X increase detected in previous work, and also a delay in the time at which cellular concentrations of RecA peaked. These differences may be attributable to lower UV dose used in our experiments, differences in growth medium and the lower temperature at which our assay was conducted compared to these studies (*Schmidt et al., 2016*).

Nevertheless, results from our live-cell experiments are generally consistent with these studies, revealing that the amount of RecA accumulated in cells attains a maximum at a time after triggering the SOS response that is much later than the de-repression of the *recA* promoter. During the SOS response, many cells undergo filamentation as cell division is blocked while some DNA synthesis continues (*Howard-Flanders et al., 1968*). The increase in *recA* gene expression counters the dilution in the cellular RecA concentration that is caused by the filamentation of the cells.

## RecA-GFP forms different types of aggregates in cells

Visualization of RecA-GFP localizations in cells revealed that RecA formed well-defined features both before and after DNA damage (*Figure 1—video 2* and *Figure 1D*). We observed three types of features: (1) foci that were present before DNA damage that dissolved in response to UV (*Figure 1D*, magenta arrows) (2) foci that appeared rapidly in the 20 min time window after UV exposure (*Figure 1D*, blue arrow) and (3) thread like structures that have been termed as RecA bundles (*Figure 1D*, green arrow). These foci exhibited various morphologies ranging from punctate foci to bundles. The foci became generally larger and more abundant after UV irradiation. To determine whether RecA foci formed in the absence of DNA damage are functionally distinct from those

formed during the SOS response, we set out to specifically visualize RecA*, the complex that is formed when RecA binds ssDNA and that is actively participating in repair. To that end, we investigated interaction partners of the ssDNA-RecA filament that are not endogenously present in *E. coli*. Since the MG1655 strain we use in our studies is cured of bacteriophage λ, we focused on co-opting the λ repressor to detect RecA* in cells (*Figure 1E*) (*Roberts and Roberts, 1975*).

## The monomeric C-terminal fragment of the bacteriophage λ repressor (mCI) is a probe for detecting RecA-ssDNA filaments

The bacteriophage λ repressor CI is responsible for the maintenance of lysogeny in *E. coli* infected with phage λ (*Echols and Green, 1971*). Oligomers of CI bind the operator regions in the constitutive $P_L$ and $P_R$ promoters in λ DNA and inhibit transcription from these promoters (*Ptashne et al., 1980*). In response to DNA damage, the λ repressor CI exhibits RecA*-dependent auto-proteolysis, much like the homologous proteins in bacteria, LexA and UmuD (*Burckhardt et al., 1988*; *Ferentz et al., 1997*; *Luo et al., 2001*; *Roberts and Roberts, 1975*; *Stayrook et al., 2008*; *Walker, 2001*). In this reaction, the ssDNA-RecA filament (RecA*) stabilizes a proteolysis-competent conformation of CI enabling auto-proteolysis at Ala111-Gly112 (*Ndjonka and Bell, 2006*; *Sauer et al., 1982*). This co-protease activity of the RecA* filament results in loss of lysogeny due to de-repression of transcription of *cI* and prophage induction of λ. The N-terminal DNA-binding domain of CI is dispensable for interactions with RecA*(*Gimble and Sauer, 1989*). A minimal C-terminal fragment of the λ repressor CI(101–229, A152T, P158T, K192A) (henceforth referred to as mCI, Molecular weight 14307.23 Da; *Figure 1F*) efficiently inhibits the auto-catalytic cleavage of a hyper-cleavable monomeric C-terminal fragment CI(92-229) (*Ndjonka and Bell, 2006*). Cryo-electron microscopy has revealed that the mCI binds deep in the groove of the RecA filament (*Galkin et al., 2009*).

## *In vitro* characterization of the binding of mCI to RecA-ssDNA filaments

Given the existing extensive *in vitro* characterization of mCI, we decided to further develop it as a probe for detecting RecA* in cells. To better understand the kinetics, cooperativity and affinity of mCI for RecA-ssDNA filaments, we first pursued an *in vitro* investigation of the interaction between mCI and RecA filaments. With the eventual goal of using mCI to detect RecA* filaments in live cell experiments, we made fusion constructs with fluorescent proteins tagged to the N-terminus of mCI via a 14-amino acid linker. To perform time-lapse imaging, we tagged mCI with the yellow fluorescent protein YPet, and to perform live-cell photoactivatable light microscopy (PALM), we tagged mCI with the photoactivatable red fluorescent protein PAmCherry. Untagged mCI and the two fluorescently labeled constructs, PAmCherry-mCI and YPet-mCI were purified and characterized for RecA-ssDNA binding as described below (See Materials and methods and Supplementary data for details, *Figure 2—figure supplement 1A*).

We first set out to interrogate the stability of mCI binding to RecA*. To that end, binding of the mCI constructs to ssDNA-RecA filaments was first assayed by surface plasmon resonance (SPR). We immobilized a 5′ biotinylated $(dT)_{71}$ ssDNA substrate on the surface of a streptavidin-functionalized SPR chip (*Figure 2A*) and assembled RecA-ssDNA filaments by injecting 1 µM RecA in buffer supplemented with ATP. This was followed by injection of buffer without RecA, but supplemented with ATPγS to minimize disassembly of the RecA filament on the ssDNA immobilized on the chip surface (*Figure 2—figure supplement 1B*). Next, the experiment was repeated but now introducing to preformed RecA* filaments solutions that not only contain stabilizing ATPγS, but also either untagged or fluorescently tagged mCI proteins. Scaled sensorgrams (*Figure 2B*) that are corrected for any disassembly of the ssDNA-RecA-ATPγS filament report on kinetics of interactions of mCI (or variant) with the highly stable RecA* filament (see also *Figure 2—figure supplement 1C*). These sensorgrams reveal that mCI associates with the RecA filament in a biphasic manner. Dissociation of mCI from the RecA filament was slow, with a dissociation halftime ($t_{1/2}$) of 850 s. In comparison, the fluorescently tagged constructs dissociated faster, but still slowly enough for use as a probe for the detection of RecA*. We measured a $t_{1/2} = 260$ s and 280 s for YPet-mCI and PAmCherry-mCI respectively. Under conditions where the interactions of mCI with ssDNA-RecA filaments could be readily probed, we also attempted to measure interactions of mCI with RecA filaments assembled on 60-mer dsDNA. In this case, we did not detect formation of dsDNA-RecA filaments even in the

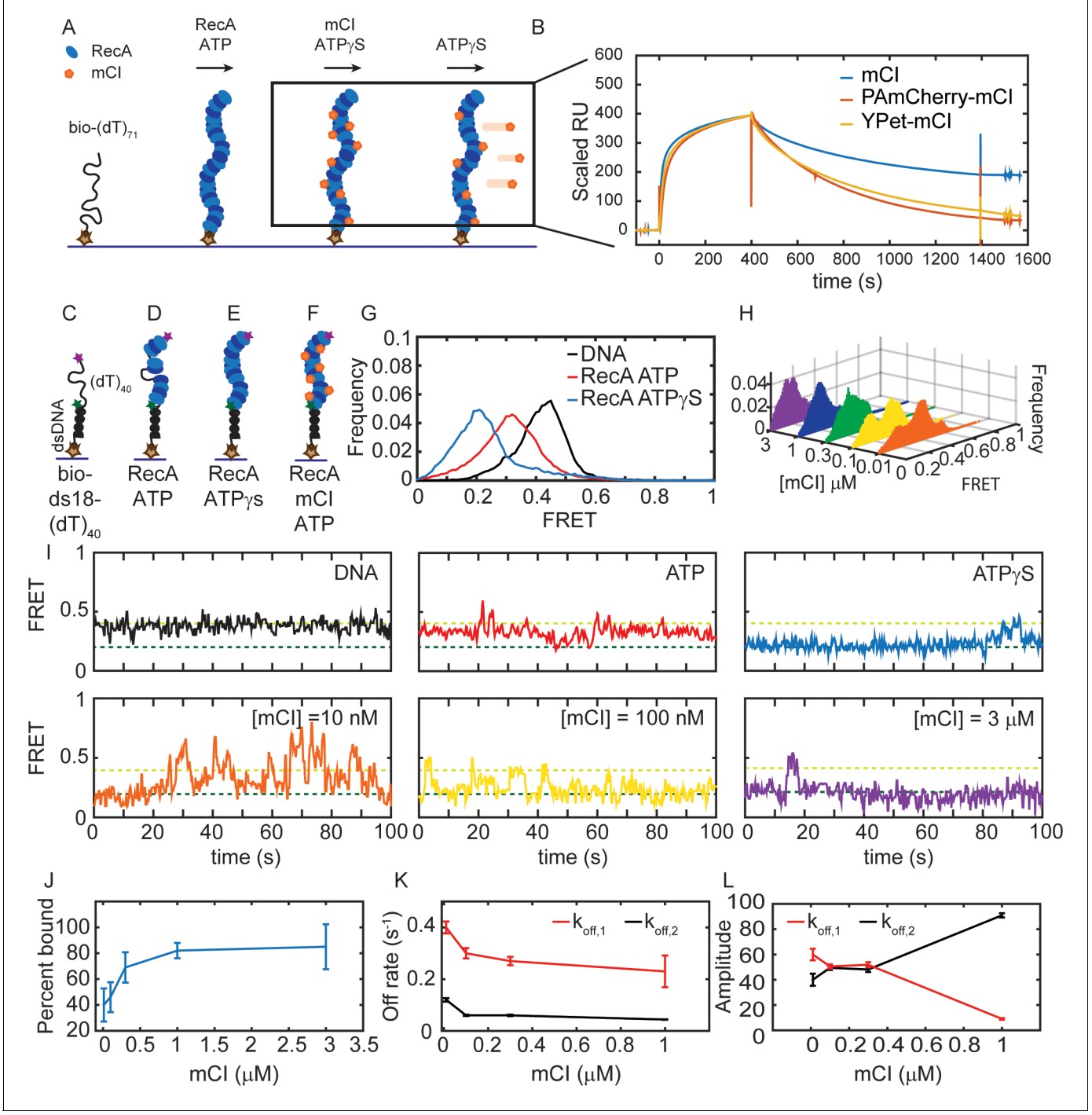

**Figure 2.** mCI stabilizes ssDNA-RecA filaments *in vitro*. (A) Schematic of SPR experiment probing association and dissociation kinetics of mCI from ssDNA-RecA-ATPγS filaments on the surface of an SPR chip. ssDNA-RecA-ATPγS filaments were assembled on a biotinylated (dT)$_{71}$ ssDNA molecule. (B) mCI (blue), YPet-mCI (yellow) or PAmCherry-mCI (red) were then flowed into the flow cell at time *t* = 0 for 400 s to monitor the association phase. Dissociation of mCI from ssDNA-RecA-ATPγS filaments was observed by leaving out mCI (or variant) from the injection buffer. Sensorgram reveals biphasic association of mCI (or variant, 1 μM) to RecA* filaments, followed by a slow dissociation from the ssDNA-RecA-ATPγS filament. Sensorgrams presented here are corrected for slow disassembly of the RecA-ATPγS filament, and data are scaled to the binding curve of YPet-mCI for purposes of comparison (see also *Figure 2—figure supplement 1C* for unscaled data). (C) Schematic of single-molecule FRET assay used to probe the influence of mCI binding on the conformational state of the ssDNA-RecA-ATP filament assembled on a ssDNA (dT)$_{40}$ overhang. Biotinylated substrate DNA (bio-ds18-(dT)$_{40}$ containing donor and acceptor fluorophores) was immobilized on a functionalized coverslip via a streptavidin-biotin interaction. (D) RecA binds the ssDNA overhang dynamically to form a ssDNA-RecA filament. (E) In the presence of ATPγS, RecA forms a stable filament. (F)

*Figure 2 continued on next page*

*Figure 2 continued*

Incubation with mCI leads to a RecA filament decorated with mCI. (G) FRET distributions observed from the substrate alone (n = 101 molecules), with RecA-ATP (1 μM RecA, 1 mM ATP, n = 179 molecules) and RecA-ATPγS (1 mM ATPγS, n = 87 molecules) from at least three independent experiments. (H) Titration of mCI shifts the RecA-ATP distribution to that of the active filament. (I) Example FRET traces of DNA substrate alone or when bound to RecA in the presence of ATPγS, or when bound to RecA in the presence of ATP and mCI (0, 10, 100, 300, 1000 and 3000 nM mCI; n = 179, 139, 77, 70, 172, 68 molecules respectively from at least three independent experiments). Dashed lines represent 'bound' (FRET = 0.2 dark green) and 'unbound' (FRET = 0.4 light green) states. (J) Fitting of the Hill equation to the percentage of bound fraction as a function [mCI] reveals a $K_D$ of 36 ± 10 nM and a cooperativity of 2.4 ± 0.2. Errors represent fitting errors to the entire data set. (K) Off-rates measured from binding of mCI to ssDNA-RecA-ATP filaments (L) Percentage amplitude of the detected rate-constants as a function of [mCI] reveals enrichment of the population decaying according to the slow off-rate as a function of [mCI] (between 40–50 molecules were analyzed at each concentration; Error bars represent fitting errors). See also *Figure 2—figure supplements 1* and *2*.

DOI: https://doi.org/10.7554/eLife.42761.005

The following figure supplements are available for figure 2:

**Figure supplement 1.** Investigation of binding of mCI, YPet-mCI and PAmCherry-mCI to RecA filaments using SPR.

DOI: https://doi.org/10.7554/eLife.42761.006

**Figure supplement 2.** Characterizing the influence of mCI on catalytic properties of RecA* using *in vitro* assays.

DOI: https://doi.org/10.7554/eLife.42761.007

presence of ATPγS, and hence could not quantify interactions with mCI or tagged variant. It is conceivable that the mCI probe can potentially interact with dsDNA-RecA filaments if they adopt a conformation similar to that of RecA*.

EM studies of RecA-ssDNA filaments have revealed that the pitch of the filament depends on the co-factor bound to it (*Egelman and Stasiak, 1986*; *Egelman and Stasiak, 1988*). Notably, ATPγS promotes the formation of the extended filament, whereas, the ADP bound filament exhibits the compressed state. We therefore set out to answer the question: what is the influence of mCI binding on the conformational state of RecA* filaments formed in the presence of ATP? To that end we adopted a single-molecule Förster Resonance Energy Transfer (smFRET) assay that has been previously used to demonstrate the nucleotide dependent conformational states of the RecA* filament (*Park et al., 2010*). We used a previously described DNA substrate consisting of a biotinylated 18-mer double-stranded region preceded by a 5'-$(dT)_{40}$ overhang ('bio-ds18-$(dT)_{40}$', See SI for details, *Figure 2C*) (*Park et al., 2010*). This substrate simulates the partly single-stranded and partly double-stranded nature of the DNA substrate that is thought to be generated in the context of replisomes encountering lesions *in vivo*. The ssDNA region is labelled with a Cy3 donor probe on one end and a Cy5 acceptor probe on the other so that the degree of extension of the ssDNA can be measured by FRET. The DNA substrate was immobilized on a streptavidin-coated surface in a flow cell and the Cy5 FRET signal was measured upon excitation of the Cy3 dye with a 532 nm laser (see SI for details). Consistent with previous FRET investigations of this DNA substrate (*Park et al., 2010*), the DNA substrate alone exhibited a FRET distribution with a mean value of 0.43 ± 0.07 (mean ± standard deviation of a single Gaussian fit to the data) reflecting the ability of the ssDNA overhang to entropically collapse and sample a large number of conformational states (*Figure 2C, G and I*; see 'DNA' trace). In the presence of ATP and RecA, the resulting FRET distribution exhibited a peak with a mean FRET value of 0.3 ± 0.1, consistent with the formation of a highly dynamic RecA filament undergoing simultaneous assembly and disassembly (*Figure 2D, G and I* 'ATP' trace). Upon incubating the DNA substrate with RecA in the presence of ATPγS, we observed a shift in the FRET distribution to an even lower value of 0.20 ± 0.07, reflecting the formation of a rigid, fully extended ssDNA-RecA filament (*Figure 2E, G and I* 'ATPγS'). Since ATPγS traps the RecA filament in an 'active' conformation that is capable of LexA repressor autocatalytic cleavage, we interpreted the 0.2 FRET state as corresponding to the active state (*Craig and Roberts, 1981*). Incubation of RecA with ADP revealed a broad FRET distribution similar to that obtained in the presence of ATP, reflecting unstable RecA filaments assembled on the ssDNA overhang (See *Figure 2—figure supplement 2A*).

Next, we studied the FRET displayed by the ssDNA-RecA-ATP filament while titrating in purified mCI (*Figure 2H* and *Figure 2—figure supplement 2B*) to gain insight into the influence of mCI binding on the stability of ssDNA-RecA-ATP filaments (*Figure 2F*). In the presence of mCI the FRET substrate exhibited a bi-modal behavior: either molecules exhibited the 0.43 FRET state or the 0.2

FRET state. Upon increasing mCI concentration, the FRET distribution gradually shifted from a mean of 0.43 to 0.20 in response to higher concentrations of mCI (*Figure 2H*). By fitting the distributions to a sum of two Gaussian fits reflecting the 'bound' state (0.20 FRET) and 'unbound' state (0.43 FRET), we were able to obtain the bound fraction at every concentration of mCI tested (*Figure 2H and J*). Fitting these data to the Hill equation yielded an equilibrium dissociation constant of 36 ± 10 nM with a Hill coefficient of 2.4 ± 0.2 (*Figure 2J*; error bars represent fitting errors). The increase in the population of molecules in the lowest FRET state in response to an increase in mCI concentration demonstrates that mCI stabilizes the RecA filament in the active conformation.

Examination of the FRET traces revealed that in the presence of mCI, the DNA substrate exhibits stochastic transitions from the RecA-bound to the unbound state (e.g. *Figure 2I* for [mCI] = 10 nM). The frequency of these transitions to the unbound state decreased in the presence of high concentration of mCI (*Figure 2I*, see also *Figure 2—figure supplement 2B*). FRET traces of DNA substrates in the presence of RecA and saturating concentrations of mCI (3 μM) exhibited stable, long-lived binding events at a FRET value of 0.20 over the time scale of acquisition (*Figure 2I*). To obtain off rates from the data, we applied a threshold of 0.3 (*Figure 2—figure supplement 2C*) to segment the trajectories such that segments with FRET values less than 0.3 were considered to reflect the 'bound' state, and those above 0.3 were considered to be the 'unbound' state. The cumulative residence time distributions for the binding events (low FRET values) in the FRET trajectories were best fit by a sum of two exponentials decaying according to a fast off rate $k_{off,1}$ = 0.23 ± 0.06 s$^{-1}$ and a slow off rate $k_{off,2}$ = 0.044 ± 0.002 s$^{-1}$ (*Figure 2K*). These off-rates were largely independent of the concentration of mCI (*Figure 2K*). However, strikingly, the fraction of the population decaying following the slower off rate increased from 35% in the absence of mCI to 91% in the presence of 1 μM mCI (*Figure 2L*).

Inside cells, RecA* performs three key catalytic functions: LexA cleavage, ATP hydrolysis and strand-exchange in its various roles in SOS induction, filament formation and DNA recombination. First, we investigated whether mCI inhibits these catalytic activities of RecA* *in vitro*. To that end, we measured the influence of mCI binding on the ATPase activity of RecA*. Incubation of preformed RecA* filaments on circular ssDNA M13mp18 substrates with micromolar concentrations of mCI revealed a pronounced inhibition of RecA* ATPase activity (*Figure 2—figure supplement 2D*). The tagged mCI variants did not significantly inhibit RecA* ATPase activity at concentrations under 500 nM (*Figure 2—figure supplement 2D*).

Current models based on EM reconstructions suggest that both mCI and LexA interact in the groove of the RecA* filament (*Galkin et al., 2009*; *Ndjonka and Bell, 2006*; *Yu and Egelman, 1993*). We next set out to investigate whether mCI could compete with, and inhibit RecA*-catalyzed cleavage of LexA. To that end, we conducted LexA cleavage assays and separated the cleavage products on a SDS-PAGE (*Figure 2—figure supplement 2E*). Quantification of the percentage of uncleaved LexA as a function of time revealed that even high concentrations of mCI did not inhibit LexA cleavage. However, μM concentrations of mCI induced a delay in the kinetics of LexA cleavage.

RecA* occupies a central role in homologous recombination (HR) where it executes the homology search and strand-exchange required for HR. We therefore investigated whether the strand-exchange activity of RecA* was influenced by mCI. We found that μM concentrations (2–10 μM) of mCI potently inhibited strand exchange (*Figure 2—figure supplement 2F*). Importantly, tagged mCI constructs did not significantly inhibit strand-exchange activity at concentrations below 500 nM (*Figure 2—figure supplement 2F*).

Taken together, these *in vitro* investigations provide insights into the consequences of mCI binding on the activity of RecA*. We found that mCI stabilizes the RecA* filament in the 'active' conformation that is capable of LexA cleavage. At high concentrations (5–10 μM), mCI can inhibit ATP hydrolysis and strand-exchange by RecA*, and delay LexA cleavage. This is consistent with mCI binding to the RecA nucleoprotein filament groove as anticipated. Importantly, at low concentrations (10–100 nM) similar to those we eventually employed as a standard *in vivo* (as described below), these key activities of RecA* are not significantly affected by the presence of mCI or tagged variant. These findings emphasize the suitability of the use of mCI derived probes for visualizing RecA* function.

## mCI inhibits SOS induction in a concentration-dependent manner

Next, we investigated whether mCI interacts with ssDNA-RecA filaments (RecA*) in cells upon DNA damage and potentially inhibits the SOS response. To that end, we created live-cell imaging vectors that express either mCI or the PAmCherry-mCI fusion from the *araBAD* promoter in a tunable manner depending on the amount of L-arabinose provided in the growth medium (*Guzman et al., 1995*). The ability of cells to induce SOS was assayed using a previously described set of SOS-reporter plasmids that express GFP in response to DNA damage (*Zaslaver et al., 2006*). In this assay, we measured the fluorescence of fast-folding GFP expressed from the *gfpmut2* gene under the SOS-inducible *sulA* promoter on a low-copy plasmid ('*sulAp-gfp*') (*Zaslaver et al., 2006*). As a control, we also measured GFP fluorescence from the promoter-less parent vector ('*gfp*'). Importantly, the copy number of these SOS-reporter plasmids is not influenced by the ultraviolet radiation (*Ronen et al., 2002*).

To measure the ability of mCI to inhibit the SOS response in cells, we co-transformed wild-type MG1655 cells with either the pBAD-mCI vector ('*mcI*'), pBAD-PAmCherry-mcI vector ('*PAmCherry-mcI*') or an empty pBAD vector ('*pBAD*'), and *sulA* reporter ('*sulAp-gfp*') or promoter-less vector ('*gfp*') to generate four strains: (1) cells that carry the empty pBAD vector and the promoter-less *gfp* vector ('*gfp* +pBAD', strain# HG257; supplemental table 2 in *Supplementary file 1*), (2) cells that carry the empty pBAD vector and the *sulA* reporter plasmid ('*sulAp-gfp* +pBAD', strain# HG258; supplemental table 2 in *Supplementary file 1*), (3) cells that carry the pBAD-mCI vector and the *sulA* reporter plasmid ('*sulAp-gfp* +mcI', strain# HG253; supplemental table 2 in *Supplementary file 1*) and (4) cells that carry the pBAD-PAmCherry-mcI vector and the SOS-reporter plasmid ('*sulAp-gfp* +PAmCherry-mcI', strain# HG285; supplemental table 2 in *Supplementary file 1*).

We then acquired time-lapse movies of these cells to observe the evolution of the SOS response over 3 hr after UV damage (*Figure 3A* and *Figure 3—video 1*). As expected, when cells carrying the *sulA* reporter plasmid and the empty pBAD vector ('*sulAp-gfp* +pBAD') were irradiated with a 20 $Jm^{-2}$ dose of UV, we observed a robust increase in GFP fluorescence (*Figure 3A*; strain# HG258; supplemental table 2 in *Supplementary file 1*). In contrast, cells carrying the promoter-less control vector and the empty pBAD vector ('*gfp* +pBAD') vectors did not exhibit any increase in GFP fluorescence in response to UV (*Figure 3A*, summarized in *Figure 3C*; *Figure 3—video 1*, strain# HG257; supplemental table 2 in *Supplementary file 1*).

After these experiments confirming the robustness of the *sulA* reporter as a readout for SOS induction, we tested whether mCI inhibits SOS induction. To that end, we grew cells carrying both the *sulA* reporter and the mCI vectors ('*sulAp-gfp* +mcI', strain# HG253; supplemental table 2 in *Supplementary file 1*) in imaging medium containing 0, $10^{-3}$ or $10^{-2}$% L-arabinose and immobilized them in flow cells. In this L-arabinose concentration regime, we expect the mCI copy number to be approximately 20, 50 and 500 copies per cell, respectively (*Ghodke et al., 2016*). As before, we quantified the cellular fluorescence at the cellular level at 5 min intervals after UV. Time-lapse acquisition after UV irradiation revealed that SOS induction was sensitive to the presence of mCI. Even leaky expression of mCI caused a measurable delay in GFP fluorescence (*Figure 3A*). This delay was found to be proportional to the expression level of mCI, and cells grown in $10^{-2}$% L-arabinose exhibited nearly complete inhibition of SOS induction during the experimental timeline of three hours after UV irradiation (*Figure 3A and C*). These data suggest that mCI competes with LexA in cells at sites of RecA* in response to DNA damage.

We then tested whether tagged mCI also similarly inhibited SOS induction. We measured GFP fluorescence in time-lapse experiments of wild-type cells carrying the pBAD-PAmCherry-mcI vector and the *sulA* reporter plasmid ('*sulAp-gfp* +PAmCherry-mcI', strain# HG285; supplemental table 2 in *Supplementary file 1*) to measure the influence of PAmCherry-mCI on SOS induction at sites of RecA* in cells. As before, we detected a similar delay in SOS induction depending on the concentration of L-arabinose in the growth medium (*Figure 3B*). At low levels of L-arabinose supplementation (<$10^{-3}$%), cells exhibit SOS induction levels that are comparable to wild-type cells. Notably, despite the weaker affinity of tagged mCI to RecA* compared to untagged mCI, their effects on SOS induction in cells were comparable. Based on these results, we chose to supplement growth medium with $5 \times 10^{-4}$% L-arabinose in further experiments aimed at visualizing RecA* filaments in cells as described below.

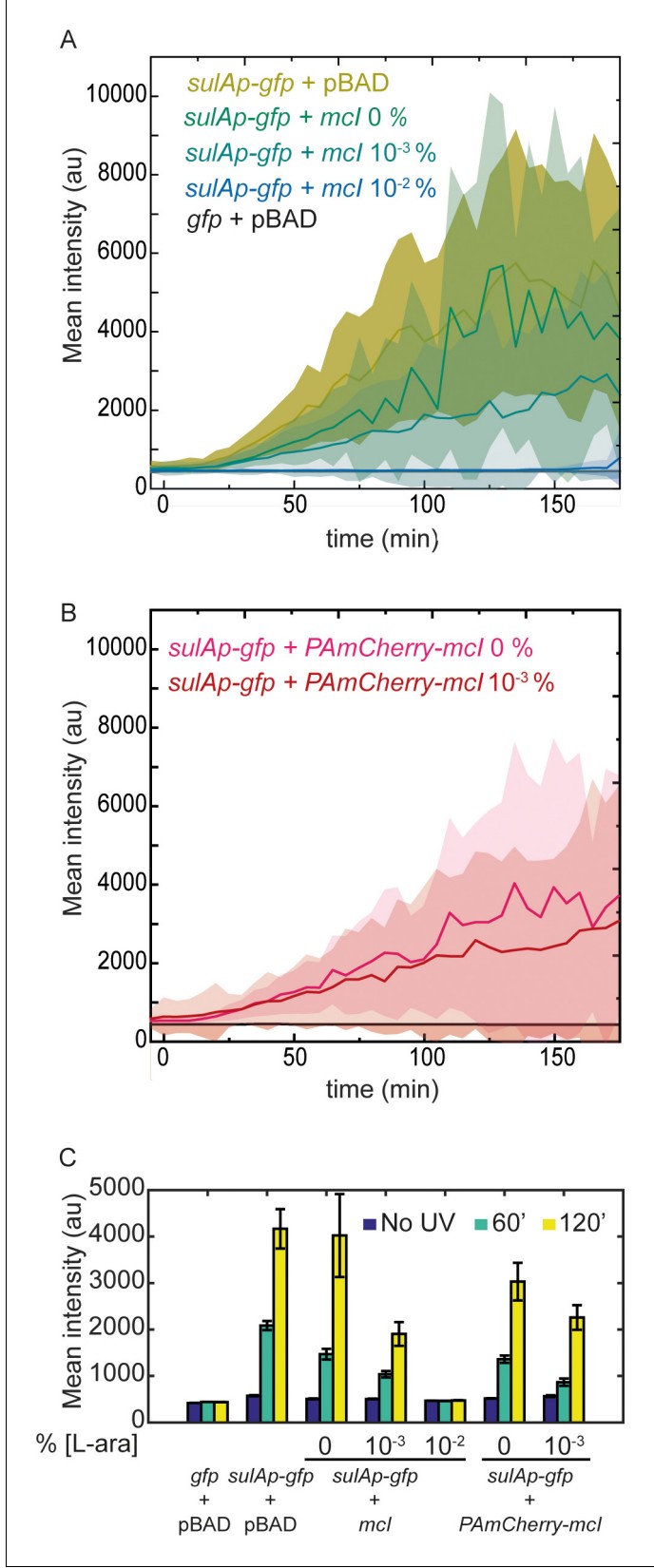

**Figure 3.** mCI inhibits the SOS response in a concentration-dependent manner. (**A**) Time-lapse experiments were performed on MG1655 cells carrying the SOS-reporter plasmids ('*gfp*' or '*sulAp-gfp*') and pBAD-mCI plasmid ('*mcI*') following irradiation with 20 Jm$^{-2}$ of UV-irradiation at time $t$ = 0 min. Mean intensity of GFP fluorescence

*Figure 3 continued on next page*

*Figure 3 continued*

was measured in cells carrying the reporter plasmid and mCI or empty vector, and plotted here as follows: 'sulAp-gfp +pBAD' cells (yellow; strain# HG258), 'gfp +pBAD' cells (black; strain# HG257), 'sulAp-gfp +mcl' (strain# HG253) (0% L-ara) (green), $10^{-3}$% L-ara (blue) and $10^{-2}$% L-ara (purple), respectively. (B) Mean intensity of GFP fluorescence in cells carrying the reporter plasmid and pBAD-PAmCherry-mcl plasmid ('sulAp-gfp +PAmCherry-mcl') (strain# HG285; 0% L-ara (pink) and $10^{-3}$% L-ara (red)) is plotted as a function of time. Shaded error bars indicate standard deviation of cellular fluorescence for all cells imaged at the indicated time point. Standard deviation was plotted to emphasize the variation of the cellular fluorescence across the population. In these experiments, 10–200 cells were analyzed from 12 fields of view at each of the 37 time points, from one independent repeat for each experimental condition. (C) Bar plots summarizing data presented in B and C under the indicated conditions at a time point before UV irradiation, one at 60, and one at 120 min after UV. Here, error bars represent standard error of the mean cellular fluorescence for all cells imaged at the indicated time point.
DOI: https://doi.org/10.7554/eLife.42761.008
The following video is available for figure 3:

**Figure 3—video 1** mCI concentration dependent inhibition of the SOS response.
DOI: https://doi.org/10.7554/eLife.42761.009

## Most RecA filaments appear at sites distal to replisomes after DNA damage

A long-standing model for SOS induction predicts that RecA* filaments are formed on chromosomal DNA when replisomes encounter UV lesions (*Sassanfar and Roberts, 1990*). These RecA* filaments are believed to be the sites of SOS induction. While several lines of evidence support the model that RecA* filaments are formed after UV irradiation, direct visualization in living *E. coli* cells has not been demonstrated. Having demonstrated that mCI can interact with RecA* in cells, we then set out to identify whether RecA* filaments form at replisomes after UV damage.

We first created a two-color strain that expresses a chromosomal *YPet* fusion of the *dnaQ* gene (that encodes the replisomal protein ε, a subunit of the replicative DNA polymerase III), and PAm-Cherry-mCI from the pBAD-PAmCherry-mcl plasmid in the presence of L-arabinose (strain# HG267; supplemental table 2 in *Supplementary file 1*). The YPet fusion has previously been shown to minimally affect the function of ε (*Reyes-Lamothe et al., 2010*; *Robinson et al., 2015*). The two-color strain was grown in medium containing small amounts of L-arabinose ($5 \times 10^{-4}$%%) to induce low expression of PAmCherry-mCI in the 10–100 nM concentration range in cells.

Next, we tested the ability of this two color strain to withstand UV exposure compared to wild-type MG1655. To that end, we performed a head-to-head comparison of UV-survival of MG1655/pBAD-mycHisB (strain# HG116; supplemental table 2 in *Supplementary file 1*) and HG267 (*dnaQ-YPet/*pBAD-PAmCherry-mcl) either in the absence of L-arabinose in the medium or when induced with $10^{-3}$% L-ara. Quantification of colony forming units revealed that the fitness of HG267 was indistinguishable from that of HG116 at the doses tested in this assay (*Figure 4—figure supplement 1A*).

Having established that HG267 behaves like wild-type in a UV-survival assay, we then set out to visualize the localization of PAmCherry-mCI with respect to the replisome in cells. This was achieved by performing time-lapse imaging on immobilized HG267 cells (5 min intervals for 3 hr after 20 Jm$^{-2}$ of UV; 30 fields of view per time point) in flow-cells in the presence of continuous flow of EZ-glycerol growth medium. In this experiment, we performed live-cell PALM to detect PAmCherry-mCI, and highly-inclined laminated optical sheet microscopy (HiLo) imaging to detect replisomes (see Materials and methods and Supplementary data for technical details related to imaging, *Figure 4—figure supplement 1B*). This approach enabled us to visualize replisomes as well as mCI foci (*Figure 4A*). To compare cells in the presence and absence of SOS induction, we repeated these experiments in cells carrying the *lexA3*(Ind⁻) allele (strain# HG311; *Figure 4—figure supplement 1C*; supplemental table 2 in *Supplementary file 1*). These cells express a non-cleavable mutant of the LexA repressor (G85D) that binds RecA*, but fails to induce SOS (*Figure 4B,E and F*) (*Markham et al., 1981*).

Next, we asked whether UV irradiation led to an increase in the number of RecA* foci in *lexA*⁺ cells. To that end, we quantified the data along three dimensions: (1) number of cells in the population exhibiting mCI (RecA*) localizations (2) numbers of replisomes and mCI (RecA*) localizations observed in each cell (3) bi-directional co-localization measurements specifically, the number of

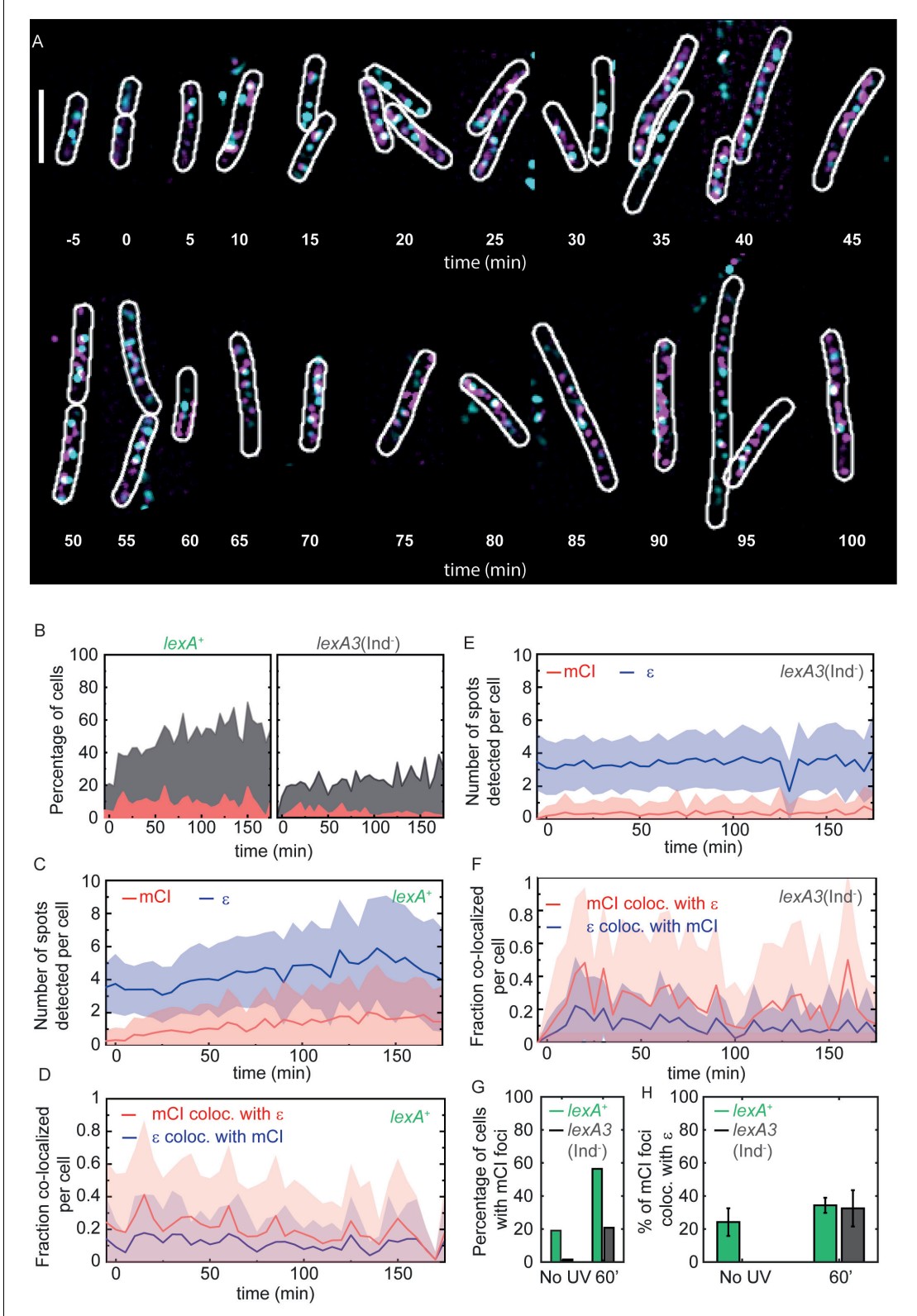

**Figure 4.** mCI co-localization with the replisome after UV irradiation. (**A**) MG1655 cells carrying the ε-YPet replisome marker (cyan) and expressing PAmCherry-mCI (magenta) from the pBAD-*PAmCherry-mcI* plasmid (strain# HG267) were grown in the presence of 5 × 10⁻⁴% L-arabinose and irradiated with 20 Jm⁻² of UV-irradiation followed by imaging for three hours. Examples of *lexA⁺* (strain# HG267) provided at indicated time points (**B**) The percentage of cells (light gray) imaged at each time point is shown for *lexA⁺* (N = 4 independent repeats) and *lexA3*(Ind⁻) (N = 3 independent

*Figure 4 continued on next page*

*Figure 4 continued*

repeats) cells. Of these, the percentage of cells exhibiting replisome as well as, mCI foci is indicated in dark gray. Red area indicates the number of cells in the population where mCI is co-localized with replisomes. Total cell counts for each time point are presented in *Figure 4—figure supplement 1D*. Number of replisome foci and PAmCherry foci were counted for each time point per cell for (C) *lexA*$^+$ and (D) *lexA3*(Ind$^-$) cells from the pooled data set. In cells exhibiting at least one replisome focus and one PAmCherry-mCI focus, the fraction of replisomes co-localizing with PAmCherry-mCI was determined (blue) and the fraction of PAmCherry-mCI co-localizing with replisomes was determined (red) for (E) *lexA*$^+$ and (F) *lexA3*(Ind$^-$) cells. (G) Bar plots summarizing percentage of cells exhibiting at least one mCI focus for *lexA*$^+$ (green) and *lexA3*(Ind$^-$) (gray) cells before UV and at 60 min after UV irradiation. (H) Bar plots summarizing extent of co-localization of ε-YPet and PAmCherry-mCI in cells with at least one mCI and ε focus. Data are presented as mean ± SEM calculated at each time point. 25–150 cells were analyzed for each time point. See also *Figure 4—figure supplement 1*.
DOI: https://doi.org/10.7554/eLife.42761.010

The following figure supplement is available for figure 4:

**Figure supplement 1.** Measurement of co-localization of mCI with the replisome.
DOI: https://doi.org/10.7554/eLife.42761.011

replisomes co-localized with RecA* and number of RecA* colocalized with replisomes in cells exhibiting both foci.

In the absence of DNA damage, 19% of all *lexA*$^+$ cells exhibited at least one PAmCherry-mCI focus (*Figure 4A and B* (*lexA*$^+$ panel, gray shaded area), $t = -5$ min; summarized in *Figure 4H*).

In response to UV damage inflicted at $t = 0$ min, we detected an increase in the number of *lexA*$^+$ cells with at least one PAmCherry focus to 56% of the population at $t = 60$ min after UV (*Figure 4B*, total numbers for each time point are presented in *Figure 4—figure supplement 1D*).

The number of replisome foci detected per cell was found to remain relatively constant ranging from 3.5 ± 1.4 (mean ± standard deviation) in the absence of UV to 4.2 ± 2.1 per cell at $t = 60$ min after UV (blue line, *Figure 4C*). The number of PAmCherry-mCI foci marking sites of RecA* was found to increase approximately five-fold from 0.3 ± 0.6 per cell before UV irradiation to 1.4 ± 0.25 per cell at $t = 60$ min (red line, *Figure 4C*).

Among cells exhibiting at least one focus (gray area), 24 ± 37% (mean ± standard deviation) of PAmCherry-mCI foci co-localized with replisomes before UV irradiation, and this number increased to 34 ± 37% at 60 min after UV (*Figure 4D and H*).

We next repeated these measurements in *lexA3* cells that are incapable of inducing the SOS response. Only 1.5% of the *lexA3*(Ind$^-$) (gray area *Figure 4B*, *Figure 4—figure supplement 1E*) population exhibited at least one PAmCherry-mCI focus (compared to 19% for wild-type) (summarized in *Figure 4H*) in the absence of UV. At 60 min after UV irradiation, only 21% of the population exhibited PAmCherry-mCI foci compared to 56% in case of the wild-type (*Figure 4B* compare gray areas, see also 4H). Quantification of the number of foci per cell revealed that the number of PAmCherry-mCI foci detected remained consistently low, starting at 0.03 ± 0.25 (mean ± standard deviation) prior to UV irradiation and reaching a value of 0.3 ± 0.7 foci per cell at 60 min (compare red lines in *Figure 4E and F*). The consistently lower number of localizations suggests that the inability to cleave LexA results in an absence of available binding sites for PAmCherry-mCI. This observation suggests that the non-cleavable LexA(G85D) protein competes with mCI for the same substrates *in vivo*.

Strikingly, *lexA3*(Ind$^-$) cells did not exhibit PAmCherry-mCI foci that co-localized with replisomes before UV irradiation (*Figure 4F*). At 60 min, 32 ± 11% of the PAmCherry-mCI foci co-localized with replisomes (*Figure 4F and H*).

Notably, RecA* could be detected even in the absence of DNA damage in *lexA*$^+$ cells. In these cells, 14 ± 25% of the replisomes exhibited co-localization with mCI (*Figure 4E*, blue curve, No UV time point). These RecA* filaments that are formed at sites of replisomes in the absence of UV light might reflect replication forks engaged in recombination-dependent DNA restart pathways or replication forks stalled at sites of bulky endogenous DNA damage.

Surprisingly, both in the case of wild-type as well as *lexA3*(Ind-) cells, the co-localization of replisomes with RecA* remained consistently low. This measured co-localization was significantly greater than co-localization by chance which remained less than 1% (see Materials and methods, *Figure 4—figure supplement 1F and G*). These results are consistent with the model that some RecA* filaments are formed in cells in the vicinity of replisomes when cells are exposed to UV light. Most RecA* filaments appear at locations distal to the replisome.

## RecA forms bundles that are stained by mCI in cells

Irradiation of RecA-GFP cells with a pulse of 20 Jm$^{-2}$ of UV led to the formation of large RecA structures during the SOS response (*Figure 5A* and *Figure 5—video 1*). These structures evolved from RecA foci into large cell-spanning structures. Previous studies have also noted the formation of these large macromolecular assemblies of RecA in response to double-strand breaks in cells (*Lesterlin et al., 2014*). These aggregates of RecA-GFP have been termed 'bundles' (*Figures 1C* and *5A* and references (*Lesterlin et al., 2014*; *Rajendram et al., 2015*)). Although these RecA bundles have been proposed to contain DNA, the nature of the RecA-DNA complex remains elusive.

We wondered whether mCI stains these RecA bundles. Examination of images of HG267 (*dnaQ-YPet*/pBAD-*PAmCherry-mcI*) cells taken at later time points revealed that localizations of PAmCherry-mCI (*Figure 5B*) resembled those of RecA-GFP in bundles (*Figure 5A*). We next investigated whether the observations of RecA bundles stained by mCI could be reproduced with a different fluorescent protein tag. We created a strain that expresses YPet-mCI from a pBAD plasmid (strain# HG143; supplemental table 2 in *Supplementary file 1*) and imaged these cells after exposure to UV-damage. Cells were induced with 10$^{-3}$% L-arabinose and immobilized in flow cells. Following irradiation with a dose of 20 Jm$^{-2}$ of UV, cells were imaged using a time-lapse acquisition protocol (1 acquisition every 5 min for 3 hr after UV). In this experiment, YPet-mCI initially exhibits cytosolic localization at the start of the experiment (*Figure 5C*, t = −5 min ('No UV' time point)) and reveals foci and bundles at later time points (*Figure 5C*, t = 1 hr and 2 hr).

Next, we tested whether the formation of these bundles required RecA that has wild-type functions. To that end, we imaged YPet-mCI in cells carrying the *recA1* allele (an inactive mutant, G160D) (*Bryant, 1988*). These cells did not exhibit foci or bundles that bound to YPet-mCI after UV (*Figure 5D*, strain# HG242; supplemental table 2 in *Supplementary file 1*). These data demonstrate that mCI recognizes a specific configuration of wild-type RecA on ssDNA – one that is able to co-operatively bind and hydrolyze ATP. Taken together these data suggest that RecA bundles consist of RecA in a conformation that resembles that of RecA*.

As an additional test, we investigated whether RecA bundles decorated by mCI are modulated by the UvrD helicase that disassembles RecA filaments in cells (*Centore and Sandler, 2007*; *Lestini and Michel, 2007*; *Petrova et al., 2015*; *Veaute et al., 2005*). Cells lacking UvrD have been shown to exhibit large foci formed by the RecA(R28A)-GFP mutant that does not form DNA-free aggregates (*Centore and Sandler, 2007*; *Renzette et al., 2005*). To that end, we visualized Δ*uvrD* cells (strain# HG235; supplemental table 2 in *Supplementary file 1*) expressing plasmid-based YPet-mCI in time-lapse fashion after exposure to a pulse of 20 Jm$^{-2}$ of UV light. These cells exhibited large RecA bundles and extensive cell filamentation that are hallmarks of constitutive SOS even in the absence of any external DNA damage (*Figure 5—figure supplement 1*). These observations are consistent with the hypothesis that RecA* filaments are stabilized by YPet-mCI, and that in wild-type cells UvrD may play a role in disassembling persistent RecA* filaments. These observations are qualitatively consistent with previous findings that UvrD limits RecA filament formation both *in vitro* and *in vivo*.

Taken together, these results demonstrate that: 1) RecA bundles are not only formed by RecA-GFP but also by wild-type RecA during the SOS response and 2) The ability to form RecA* – a high-affinity complex on ssDNA – is critical for the formation of RecA bundles. Further, the lack of mCI features in the *recA1* background suggest that far from being DNA-free aggregates of RecA, these bundles contain an ordered assembly of RecA that is bound to DNA.

## RecA-GFP forms storage structures in cells

Having established that the foci that appear after DNA damage correspond to RecA* intermediates, we next turned our attention to the RecA foci that disappear upon DNA damage (*Figure 1D*, magenta arrows, and *Figure 1—video 2*). RecA forms DNA-free aggregates *in vitro*, and the R28A mutant of RecA exhibits fewer foci in cells when tagged to GFP (*Eldin et al., 2000*; *Renzette et al., 2005*). We therefore hypothesized that the RecA-GFP foci that disappear after UV exposure represent storage structures. To detect storage structures of RecA in live cells, we imaged *recA-gfp* cells in the absence of DNA damage (strain# HG195; supplemental table 2 in *Supplementary file 1*). Cells exhibited punctate foci that appear to be positioned outside the nucleoid (*Figure 1D*,

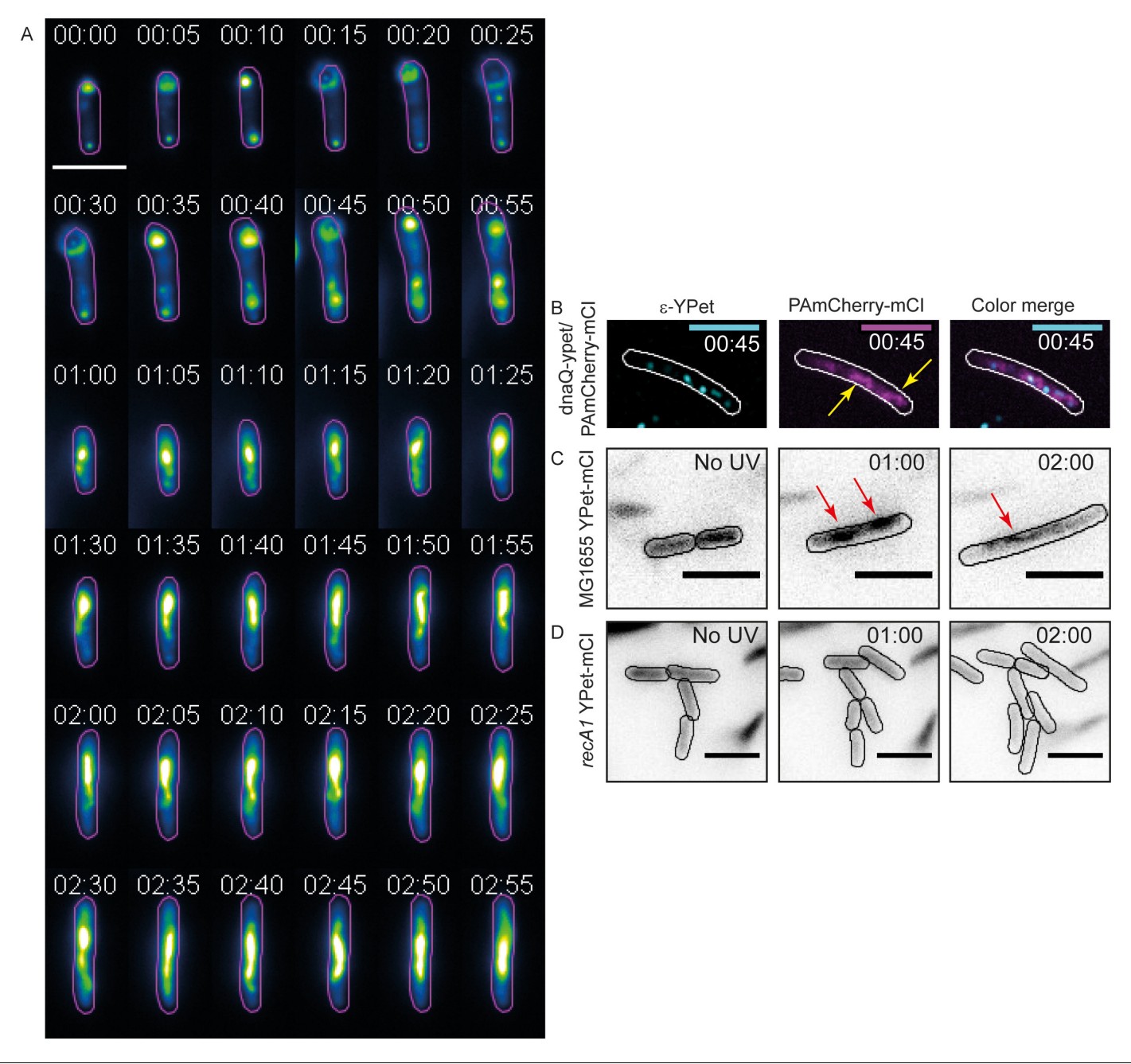

**Figure 5.** mCI stains RecA bundles after UV-damage. (**A**) Montage of a single *recA-gfp* cell exhibiting large, dynamic RecA-GFP structures at late time points from **Figure 5—video 1**. Note that the immobilized cell divides at t = 1 hr. The daughter cell is carried away by flow. Cell outlines are provided as a guide to the eye. Time is indicated as hh:mm. (**B**) At late time points in the DNA damage response, PAmCherry-mCI forms large bundles in *recA⁺ cells*. Shown here is an example of an overlay of the mCI signal (magenta) and replisomal ε foci (cyan) at *t* = 45 min after 20 Jm⁻² UV (N = 4 independent experiments). Yellow arrows point to RecA bundles. For purposes of illustration, peaks in the ε images were enhanced using a discoidal average filter. (**C**) YPet-mCI also forms bundles (indicated by red arrows) in response to UV-damage in *recA⁺* cells (N = 3 independent experiments). (**D**) Cells carrying the *recA1* allele do not exhibit foci or bundle formation upon UV-irradiation under identical conditions as in panel (n > 250 cells from 12 fields of view at each of the 37 time points). Scale bar s correspond to 5 μm. See also **Figure 5—figure supplement 1**. Cell outlines provided as a guide to the eye.

DOI: https://doi.org/10.7554/eLife.42761.012

The following video and figure supplement are available for figure 5:

**Figure supplement 1.** Detection of RecA bundles using mCI in ΔuvrD cells.

*Figure 5 continued on next page*

*Figure 5 continued*

DOI: https://doi.org/10.7554/eLife.42761.013

**Figure 5—video 1.** Visualization of RecA-GFP bundles in *recA-gfp* cells after UV exposure.

DOI: https://doi.org/10.7554/eLife.42761.014

*Figure 1—video 2* and *Figure 6A*). Notably, cells carrying these structures did not exhibit markers of distress, namely cell filamentation.

We reasoned that storage structures of RecA would need to satisfy two criteria to be distinguished from complexes active in DNA repair and from polar aggregates representing mis-folded proteins: (1) the size or number of these structures should be proportional to the amount of RecA present in the cell and (2) RecA stored in these structures should be available for biological function when required, that is, after DNA damage. Therefore, we set out to investigate whether the RecA structures observed in the absence of DNA damage exhibited behavior consistent with these expectations.

## Size of RecA storage structures is determined by the amount of cellular RecA

First, we tested if the size of these punctate foci is dependent on the copy number of RecA in cells. To that end, we pursued a strategy involving over-expression of unlabeled RecA from a plasmid in *recA-gfp* cells and measuring whether the foci become larger as the amount of untagged RecA increases and integrates into the structures with the labelled RecA. We chose to express wild-type untagged RecA instead of tagged RecA for two reasons: (a) as evidenced by the three different types of structures observed in *recA-gfp* cells, unambiguous detection of storage forms of RecA as opposed to DNA-bound forms is very difficult when cells carry only the RecA-GFP fusion (*Figure 1D* and *Figure 1—video 2*). (b) Since *the recA-gfp* strain exhibits compromised UV-survival and recombination, we cannot be guaranteed that the structures detected in time-lapse experiments of this strain are authentic. Expression of excess wild-type RecA alleviates these issues. Since wild-type RecA can out-compete tagged RecA in catalytic functions, the RecA-GFP proteins would then serve as markers for DNA-free complexes of RecA.

To unambiguously observe storage forms of RecA, we created plasmids that express wild type RecA protein at two different levels: first, a low-copy plasmid that expresses *recA* from the constitutive *recAo281* operator (pConst-*recA*; see SI for details) (*Uhlin et al., 1982*; *Volkert et al., 1976*) and a second version of that plasmid (pG353C-*recA*) where expression was cut in half upon incorporating an altered ribosome binding site (RBS) (see *Figure 6—figure supplement 1A*). We then imaged *recA-gfp* cells carrying one or the other of these plasmids.

Time-lapsed imaging of undamaged cells (5 min intervals for 3 hr) revealed that the RecA-GFP signal was confined to a single large feature (*Figure 6A*). To quantify the size of RecA features in the absence of DNA damage, we measured the maximum Feret diameter (referring to the largest physical dimension of the structure; *Figure 6A*) of the feature at a threshold above the background (*Figure 6B*, see SI for details). Comparison of the Feret diameters of features in *recA-gfp* cells carrying the pConst-*recA* and pG353C-*recA* vectors revealed a strong dependence on the expression level of untagged wild-type RecA protein (*Figure 6B*). *recA-gfp*/pConst-*recA* cells exhibited larger features than *recA-gfp*/pG353C-*recA* cells or *recA-gfp* cells alone. Notably, the storage structures exhibited cross-sections that were circular (in the case of *recA-gfp cells)* or elliptical (in the case of *recA*-gfp/pConst-*recA* or *recA-gfp/*pG353C-*recA* cells) unlike the previously described thread-like filamentous RecA-bundles (*Kidane and Graumann, 2005*; *Lesterlin et al., 2014*). In the absence of DNA damage, these structures were stably maintained in cells in *recA-gfp*/pG353c-*recA* cells (*Figure 6—video 1*). All cells exhibited storage structures, and upon cell division, the structure was inherited by one of the daughter cells. Notably, cells did not exhibit markers of distress consistent with induction of SOS, suggesting that these storage structures do not impede DNA replication during growth, and are likely not assembled on DNA.

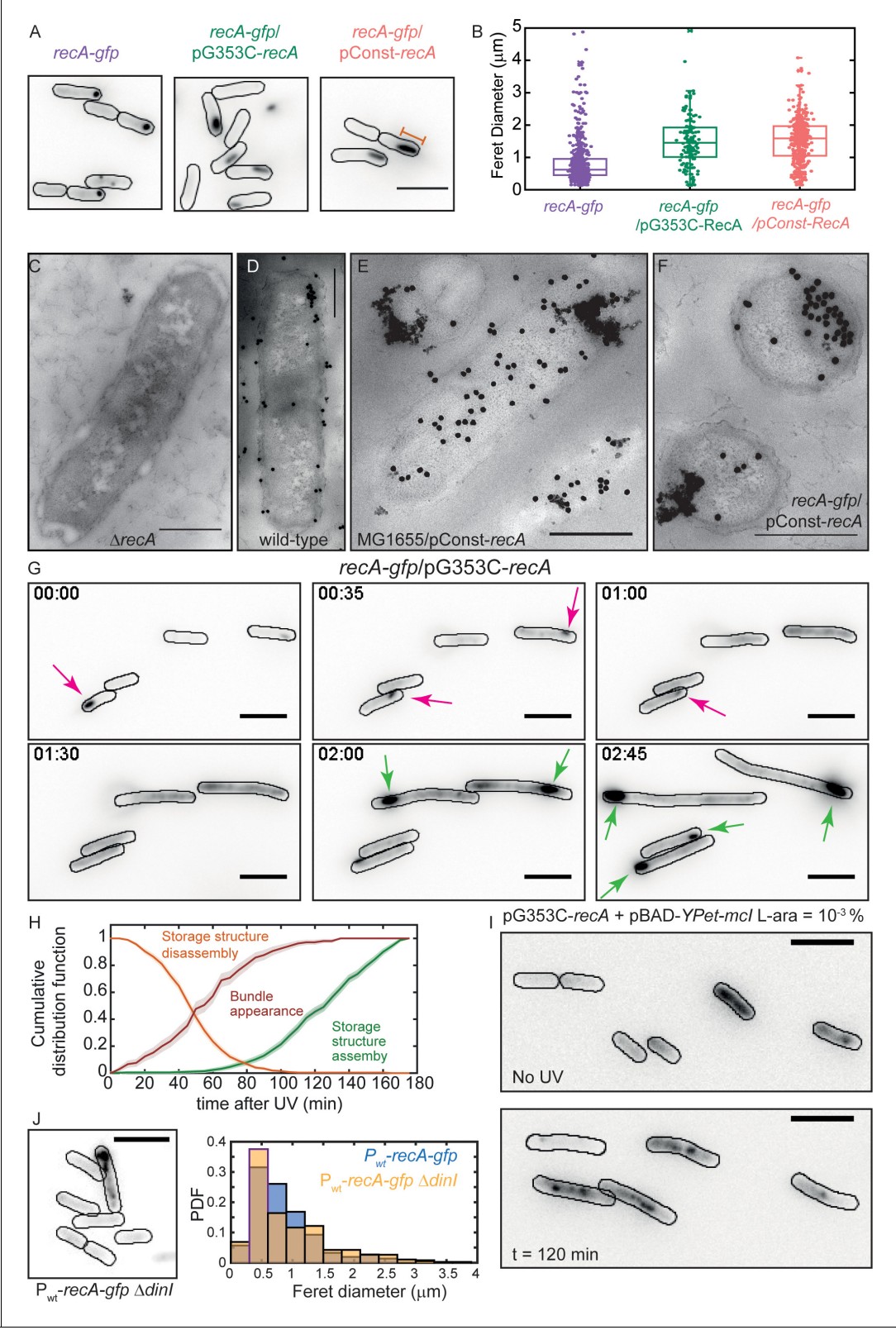

**Figure 6.** Excess RecA is stored in storage-structures. (**A**) Montage of *recA-gfp* (strain# HG195), *recA-gfp*/pG353C-*recA* (strain# HG406) and *recA-gfp*/pConst-*recA* cells (strain# HG411) imaged in the absence of UV damage. See also *Figure 6—video 1*. Scale bar corresponds to 5 µm. (**B**) Box and whiskers plot of maximum Feret diameter of storage structures in *recA-gfp* (purple; n = 528 structures), *recA-gfp*/pG353C-*recA* (green; n = 137 structures) and *recA-gfp*/pConst-*recA* cells (orange; n = 399 structures). Mean and 25th/75th percentile are encapsulated in the box. Orange bar in panel

*Figure 6 continued on next page*

*Figure 6 continued*

A represents the maximum Feret diameter for that particular storage structure. Pairwise Kolmogorov-Smirnov test to compare the distributions of the Feret diameters of the storage structures revealed statistically significant differences for *recA-gfp* vs. *recA-gfp*/pG353C-*recA* (p = $1.3327\times10^{-29}$) and *recA-gfp* vs. *recA-gfp*/pConst-*recA* (p = $1.54\times10^{-60}$), rejecting the null hypothesis that the measurements for the two strains arise from the same distribution. Electron microscopy images of (C) Δ*recA* (D) wild-type *recA* (E) MG1655/pConst-*recA* and (F) *recA-gfp*/pConst-*recA* cells stained with gold nanoparticles labelled with RecA antibody. Note the appearance of aggregates of gold nanoparticles in panel E at locations consistent with those observed in panel A for *recA-gfp*/pConst-*recA* cells. Untagged, over-expressed RecA reveals gold nanoparticle localizations consistent with those expected from RecA storage structures. Scale bar corresponds to 1 μm. (G) Montage of frames from a time-lapse experiment of *recA-gfp*/pG353C-*recA* cells exposed to UV (see also *Figure 6—video 2*). RecA forms storage structures in the absence of DNA damage (0 min) in cells. Storage structures dynamically dissolve after DNA damage (1 hr; magenta arrows). Storage structures reform by sequestering excess RecA synthesized during SOS after repair (2 hr and 2 hr 45 min time points; green arrows). N = 5 independent experiments. (H) Cumulative probability distributions of time of solubilization of storage structure (yellow) and time of appearance (light green) of storage structures from *recA-gfp*/pG353C-*recA* (strain# HG406) cells (N = 4 independent experiments). Red line represents cumulative distribution function of time of first incidence of RecA bundles in *recA-gfp* cells (strain# HG195, n = 108 bundles). Shaded error bars represent standard deviation of the bootstrap distribution obtained by sampling 80% of the data 1000 times. In each case, 100–150 cells were analyzed that were present for the duration of observation (3 hr). (I) YPet-mCI does not stain storage structures in MG1655/pG353C-*recA* pBAD-*YPet-mcI* cells (strain# HG446) in the absence of DNA damage, but forms features after UV damage (shown here is a still at 120 min). Cell outlines provided as a guide to the eye. N = 2 independent experiments. See also *Figure 6—video 3*, and *Figure 6—figure supplement 1*. Scale bar represents 5 μm. (J) MG1655 cells carrying the *recA-gfp* fusion under the native *recA* promoter and Δ*dinI* (strain# EAW767) exhibit fewer storage structures than *dinI*⁺ cells. On average, 27% of EAW767 ($P_{wt}$-*recA-gfp* ΔdinI) cells exhibited structures (mean Feret diameter = 0.9 ± 0.6 μm, n = 702 cells), compared to 43% of EAW428 ($P_{wt}$-*recA-gfp*) cells (mean Feret diameter = 0.9 ± 0.5 μm, n = 855 cells). N = 2 independent experiments. Scale bar represents 5 μm. See also *Figure 6—figure supplement 1*. A Kolmogorov-Smirnov test to compare the two distributions did not reject the null hypothesis that the two measurements of the Feret diameters for the two strains arose from the same distribution, resulting in a p-value of 0.15.

DOI: https://doi.org/10.7554/eLife.42761.015

The following video and figure supplement are available for figure 6:

**Figure supplement 1.** Detection of storage structures of RecA.
DOI: https://doi.org/10.7554/eLife.42761.016
**Figure 6—video 1.** Time-lapse imaging of *recA-gfp*/pG353C-*recA* cells.
DOI: https://doi.org/10.7554/eLife.42761.017
**Figure 6—video 2.** Time-lapse imaging of *recA-gfp*/pG353C-*recA* cells after UV damage.
DOI: https://doi.org/10.7554/eLife.42761.018
**Figure 6—video 3.** Time-lapse imaging of YPet-mCI in cells over-expressing wild-type RecA.
DOI: https://doi.org/10.7554/eLife.42761.019

## Untagged RecA also forms storage structures

To date, *in vivo* studies on storage structures of RecA have relied on visualization of GFP-tagged fusions of RecA. We examined whether the ability of RecA-GFP to assemble into storage structures was a property shared by untagged RecA. To that end, we collected electron-microscopy images of cells stained with immunogold-labeled anti-RecA antibody in four genetic backgrounds: (1) Δ*recA*; (2) wild-type MG1655; (3) MG1655/pConst-*recA*; and (4) *recA-gfp*/pConst-*recA* (*Figure 6* panels C-F, respectively). As expected, clusters of gold-labelled RecA antibodies could be observed in all samples except Δ*recA* cells (*Figure 6C*). Cells carrying the RecA over-expresser plasmid pConst-*recA* exhibited strong RecA staining that was localized to the membrane (*Figure 6E and F*. See *Figure 6—figure supplement 1B* for additional examples). These results support the conclusion that excess RecA is stored in the form of membrane-associated, phase separated, storage structures even in cells carrying untagged, wild-type RecA. Further, these results demonstrate the suitability of the use of RecA-GFP as a marker for studying the localization of RecA.

## Storage structures of RecA dissolve after DNA damage

That polar assemblies of RecA-GFP represent storage structures has been a foregone conclusion in the literature. Certainly, wild-type RecA can self-assemble *in vitro* and *in vivo* under certain experimental conditions, but whether these structures indeed contain RecA in a 'stored' form, in a manner where it is available for SOS functions has remain unaddressed. We therefore interrogated whether the stored RecA was available to support repair during the SOS response. To that end, we exposed *recA-gfp*/pG353C-*recA* cells to UV radiation and monitored the dynamics of the storage structures in time-lapsed fashion. We found that the storage structures dissolve within one hour after

introducing damage, flooding the cell with RecA-GFP (*Figure 6G*, *t* = 1 hr). At later time points, the storage structures re-appeared at locations close to the poles (*Figure 6G*, *t* = 2 hr 45 min), suggesting that the RecA is stored away until needed (see also *Figure 6—video 2*).

We quantified the dynamics of storage structure disassembly by plotting the cumulative distribution function of loss of storage structures for the population of cells that possessed a distinct storage structure as a function of time (*Figure 6H*; orange curve). We found that for *recA-gfp*/pG353C-*recA* cells, half of the storage structures were lost within 45 ± 5 min after UV damage (see Materials and methods for details). We then plotted the cumulative probability distribution of time of appearance of storage structures after SOS induction (*Figure 6H*, green curve). In these cells, half of the population of storage structures that formed after UV damage, did so after 135 ± 5 min after UV.

## RecA storage structures are not RecA bundles

Next, we investigated whether the RecA in storage structures adopts a conformation that is recognized by mCI. Fluorescence imaging of plasmid-based YPet-mCI expressed from the pBAD promoter in wild-type cells expressing pG353C-*recA* (strain# HG446) did not reveal any morphological features consistent with those of the storage structures observed in *recA-gfp*/pG353C-*recA* cells. YPet-mCI was found to be cytosolic in the absence of DNA damage, suggesting a lack of stable association with RecA storage structures (*Figure 6I*, 'No UV' time point). As noted earlier, mCI foci were rare in the absence of DNA damage. However, in response to UV irradiation, cytosolic YPet-mCI was found to form foci and bundles (*Figures 6I* and 120 min time point; see also *Figure 6—video 3*). Observations of cells expressing plasmid-based mCI in *recA-gfp*/pG353C-*recA* revealed no detectable influence of mCI on the morphology of these structures (*Figure 6—figure supplement 1C*), reinforcing the interpretation that mCI does not interact with storage structures of RecA in the absence of damage.

To further confirm that storage structures are indeed distinct from RecA bundles, we characterized the kinetics of RecA-GFP bundle formation in *recA-gfp* cells (strain# HG195; supplemental table 2 in *Supplementary file 1*). Upon UV irradiation, cytosolic RecA-GFP forms foci that progress to form large, cell-spanning bundles over the course of several tens of minutes (*Figure 1—video 2*), unlike RecA storage structures observed in the *recA-gfp*/pG353C-*recA* cells. Plotting a cumulative probability distribution of time of incidence of bundle formation revealed that half of all bundles in *recA-gfp* cells appear by 60 min after UV (*Figure 6H*, red curve). Curiously, this timing is consistent with measurements of incidence of bundle formation during double-strand break repair (*Lesterlin et al., 2014*).

## DinI promotes the formation of storage structures in cells

The DinI protein is a modulator of RecA function (*Lusetti et al., 2004a*; *Lusetti et al., 2004b*; *Renzette et al., 2007*). In solution, the C-terminal tail of DinI mimics ssDNA, enabling interactions with monomeric RecA (*Ramirez et al., 2000*). Since free RecA assembles to form storage structures, we next investigated whether storage of RecA was influenced by DinI. Considering that expression level of RecA influences storage structure formation, we first constructed a strain carrying the *recA-gfp* chromosomal fusion under its native wild-type promoter (P$_{wt}$-*recA-gfp*; strain# EAW428). We deleted *dinI* in this background (strain# EAW767). In the absence of DNA damage, we detected storage structures in fewer Δ*dinI* cells (27% of 702 cells) compared to *dinI*$^+$ cells (43% of 855 cells) (see *Figure 6J*). Over-expression of DinI from pBAD-*dinI* in Δ*dinI* cells further confirmed this result: cells recovered storage structures in the presence of L-arabinose (see *Figure 6—figure supplement 1D*). These findings suggest that RecA storage structure formation may be promoted by DinI.

## Discussion

In this work, we have used the C-terminal fragment of the λ repressor in conjunction with single-molecule imaging techniques in live cells to examine RecA protein dynamics in response to SOS induction. In the absence of DNA damage, we see that RecA is largely sequestered in storage structures. Upon UV irradiation, these storage structures dissolve and the cytosolic pool of RecA rapidly nucleates on DNA to form early SOS signaling complexes, followed by RecA bundle formation at later time points. Our analysis indicates that the bundles are bound to DNA in the form of RecA*. Upon completion of repair, RecA storage structures reform. Our use of the mCI reagent, which associates

with DNA-bound and activated RecA* complexes, allows us to eliminate the ambiguity associated with earlier observations utilizing RecA fusion proteins with limited functionality and for the first time provide access to the spatial and temporal behavior of the various forms of RecA structures within the cell. In addition, whereas some RecA foci that form after DNA damage co-localize with replisomes, the majority do not.

We set out to use binding partners of RecA to probe intracellular localization of SOS-signaling RecA complexes. Several proteins associated with the SOS response, notably LexA, UmuD, and the λ repressor, interact with RecA* to effect their autocatalytic cleavage. The interaction of these proteins with the activated RecA nucleoprotein filament has been a subject of intense investigation (*Cohen et al., 1981*; *Gimble and Sauer, 1985*; *Little, 1982*). Even though each of these proteins interacts with a different set of residues on the RecA* filament, they all occupy the helical groove of the RecA filament prior to auto-proteolysis (*Frank et al., 2000*; *Galkin et al., 2009*; *Yu and Egelman, 1993*). We sought to exploit this key feature by using a fluorescently labelled C-terminal fragment of λ repressor CI (denoted mCI) to visualize RecA-DNA complexes. The mCI construct binds specifically to RecA* (*Figure 2*, *Figure 2—figure supplement 1*). Binding of mCI stabilizes RecA* in the 'active' conformation capable of mediating LexA cleavage, exhibiting an equilibrium dissociation constant of $36 \pm 10$ nM and a Hill coefficient of $2.4 \pm 0.2$ for the binding of mCI to ssDNA-RecA filaments assembled on a $dT_{40}$ ssDNA overhang. Based on previous findings that one mCI contacts two RecA monomers, we estimate that up to six mCI molecules can decorate the RecA-ssDNA filament composed of up to 13 RecA monomers on the $dT_{40}$ DNA substrate under conditions of saturating mCI concentration (*Galkin et al., 2009*; *Ndjonka and Bell, 2006*). We confirmed that mCI interacts with RecA filaments in live cells by probing SOS induction after UV damage (*Figure 3*). We found that mCI has the potential to robustly inhibit SOS induction at high concentrations. SOS induction is retained, albeit delayed, at mCI concentrations employed in this study.

The leading model for SOS induction is that replication forks fail at sites of lesions and produce large tracts of ssDNA that templates nucleation of RecA filaments. Visualizing this model in cells has been challenging due to the difficulties associated with co-localization of a high-abundance protein (RecA) with a handful of replisomes. Our strategy involving fluorescently tagged mCI enabled us to examine the location of RecA* foci in nucleoids relative to the replisomes for the first time in *E coli*. We found that the average number of replisome foci did not change after DNA damage, confirming that most replisomes are not disassembled after UV (*Figure 4*). Live-cell PALM imaging of mCI revealed foci that depended on the presence of wild-type RecA and DNA damage. Surprisingly, 20% of wild-type cells exhibited RecA* foci during normal growth. However, only 24% of these co-localized with replisomes. The remaining 76% of sites of RecA* detected during normal growth did not co-localize with replisomes. Upon exposure to ultraviolet light, 56% of cells exhibited RecA* foci that were visualized by mCI, with up to 35% of the RecA* foci co-localized with replisomes at 60 min in rich media (*Figure 4*). A previous report on co-localization of RecA-GFP with DnaX-mCherry in *Bacillus subtilis* growing in minimal media reported a basal co-localization of $74.8 \pm 8.4\%$ with an increase to $84.3 \pm 5.8\%$ at 5 min after 40 $Jm^{-2}$ UV treatment (*Lenhart et al., 2014*). The extent of co-localization of RecA* and replisomes detected in our experiments, in *E. coli* cells growing in media that supports multi-fork replication is lower both before and after UV irradiation. The RecA* foci that co-localize with replisomes are likely associated with replisomes that are stalled at sites of DNA damage. We postulate that RecA* foci that are not co-localizing with replisomes are forming in DNA gaps that are formed and left behind by the replisome (*Howard-Flanders et al., 1968*; *Rupp and Howard-Flanders, 1968*; *Yeeles and Marians, 2013*). Notably, most foci of the translesion DNA polymerases IV and V also form at nucleoid locations that are distal from replisomes, both before and after SOS induction (*Henrikus et al., 2018*; *Robinson et al., 2015*).

The large cell-spanning structures termed RecA threads or bundles (we have adopted the latter term) (*Kidane and Graumann, 2005*; *Lesterlin et al., 2014*; *Rajendram et al., 2015*) that form after SOS induction deserve special mention. Following the initial phase of RecA* formation, cells expressing YPet-mCI, PAmCherry-mCI or RecA-GFP exhibited large RecA bundles. The formation of these bundles was also contingent upon the presence of wild-type RecA. The *recA1* allele failed to support focus or bundle formation, consistent with the inability of the RecA(G160D) to induce SOS and HR functions (*Bryant, 1988*). Additionally, cells lacking UvrD exhibited constitutive RecA bundles. These bundles are thus a hallmark of the DNA damage response and may have special functionality in the homology search required for recombinational DNA repair (*Lesterlin et al., 2014*). Here, we show

that the bundles bind to our mCI probe. This implies that the bundles are either bound to DNA and thus activated as RecA*, or at a minimum are in a RecA*-like conformation that permits mCI binding.

Interestingly, despite the differences in the nature of the DNA damage inflicted, the timing of RecA bundle formation in our UV experiments coincided closely with that of RecA bundles observed upon induction of site specific double-strand breaks in the chromosome (*Lesterlin et al., 2014*). The bundles may thus be nucleated by RecA binding to either gaps or resected double strand breaks. At later time points, polymerization of RecA* filaments nucleated at ssDNA gaps could extend onto dsDNA, and manifest as bundles in our experiments.

How this RecA* is organized in the RecA bundles remains unclear. Observations of *recA-gfp* cells revealed that RecA bundles are dynamic, and change in physical dimensions over time. As previously noted, RecA-GFP bundles exhibit a thick central body and thin extensions. The variation in the cross-section of the RecA bundle along the length would rule out a RecA bundle being a single extended RecA* filament. Ordered assembles of RecA have been previously detected in cells after nalidixic acid treatment (*Levin-Zaidman et al., 2000*). However, other configurations ranging from multiple, folded RecA* filaments assembled on resected double-strand breaks (*Egelman and Stasiak, 1986*) to a tangled web of pre-synaptic and synaptic complexes of RecA* on dsDNA (*Pinsince and Griffith, 1992*) may potentially manifest as RecA bundles.

RecA bundles also interact with anionic phospholipids in the inner membrane (*Rajendram et al., 2015*). Notably, UmuC also localizes primarily at the inner-membrane upon production and access to the nucleoid is regulated by the RecA* mediated UmuD cleavage (*Robinson et al., 2015*). This transition occurs late in the SOS response (after 90 min), at a time-point when most, if not all of the cells in the population exhibit RecA bundles. The origins, maturation and additional catalytic roles of RecA bundles in the SOS response require additional investigation.

Our experiments enable us to distinguish storage structures from SOS signaling complexes and RecA bundles based on three qualities: (1) Storage structures dissolve after UV damage whereas RecA bundles are formed in response to DNA damage. (2) Storage structures often exhibit a polar localization, whereas RecA-bundles form along the cell length. (3) The SOS signaling complexes and RecA bundles are visualized by binding to mCI, whereas the RecA storage structures are not. Finally, we found that DinI promotes storage structure formation: cytosolic RecA in normal growing cells was found to be sequestered in structures by simply over-expressing DinI from a plasmid.

Taken together, these data for the first time provide a full picture of a process that was first hypothesized by Story and co-workers in 1992 suggesting that RecA can undergo a phase transition to form DNA-free assemblies in live cells and redistribute into the cytosol where it becomes available for DNA-repair functions (*Figure 7*) (*Story et al., 1992*). Within a few min after encountering bulky lesions, replication forks synthesize ssDNA substrates that are rapidly coated by cytosolic RecA to form RecA*. These RecA* enable auto-proteolysis of LexA to initiate the SOS response and increase the levels of cellular RecA protein. Meanwhile, storage structures of RecA dissolve, making RecA available for biological functions. The RecA* foci elongate over several hours into elaborate bundles that may have multiple functions. Finally, excess RecA is sequestered away into storage structures approximately 2 hr after DNA damage, after DNA repair is complete and normal growth is restored.

## Materials and methods

### Construction of vectors used in this study

#### Cloning of pBAD-S

pBAD-S was created by amplifying the kanamycin cassette from pEAW507 using NtermFusion_FW and NtermFusion_R primers (*Robinson et al., 2015*). The PCR product was then digested with *NcoI/SalI* and ligated into pBAD-myc-HisB. This resulted in the insertion of a linker sequence 'S'(MSAGSAAGSGEF•) and a FRT-Kan-FRT site.

#### Cloning of pBAD-*mcI*, pBAD-*YPet-mcI* and pBAD-*PAmCherry-mcI* for live-cell imaging

Sequence verified *mcI* or *YPet-mcI* geneblocks were cloned into pBAD-S between *NotI/ApaI* using standard sub cloning protocols to yield pBAD-mCI and pBAD-*YPet-mcI* respectively. pBAD-*PAmCherry-mcI* was created by amplifying the *PAmCherry1* gene from RpoC-PAmCherry strain from the

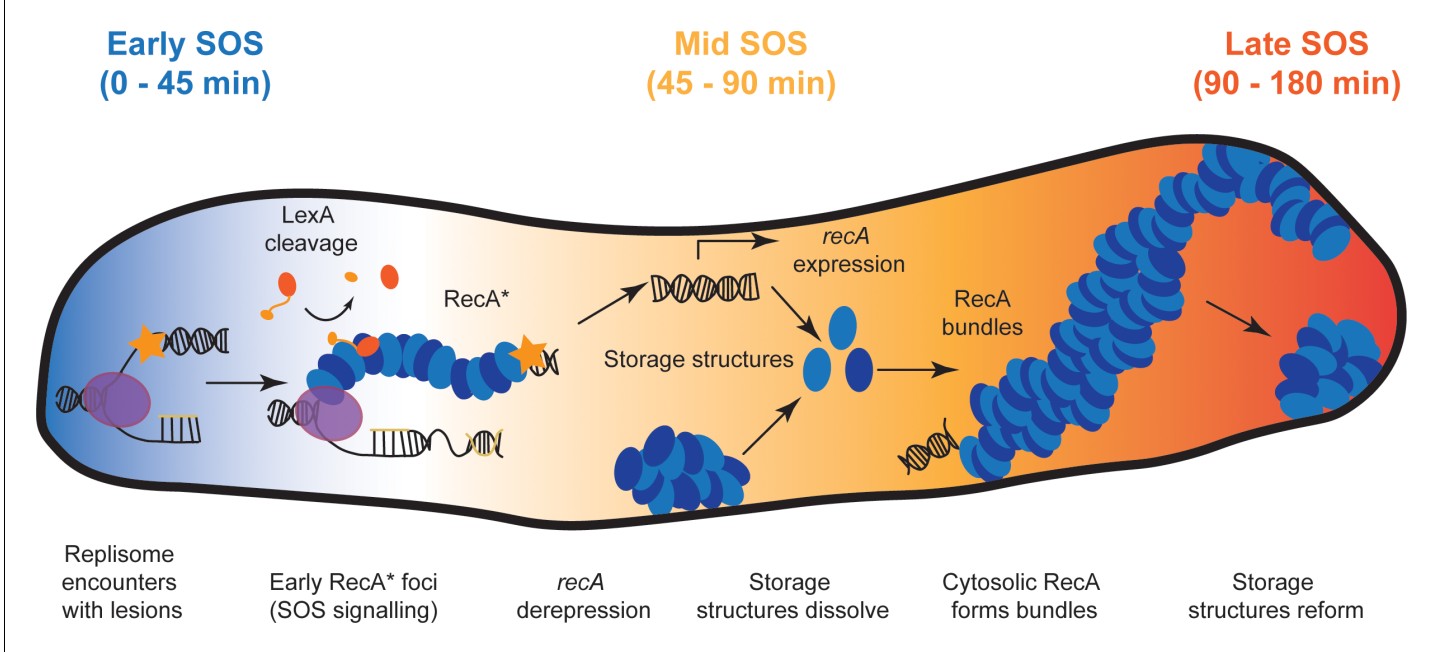

**Figure 7.** Model for organization of RecA complexes after DNA damage. The SOS response in *E. coli* is composed of three stages. Detection of UV damage leads to formation of ssDNA-RecA filaments at sites of replisomes triggering the early stages of the SOS response. In this stage, ssDNA-RecA (RecA*) filaments catalyze auto-proteolysis of LexA to induce SOS and deregulate expression of the SOS inducible genes, including *recA*. Cells stop dividing and instead begin to filament. Storage structures of RecA dissolve in response to DNA damage to make RecA available for repair and recombination in the middle stage of the SOS response (45–90 min). RecA forms membrane-associated bundles starting in mid SOS. These bundles mature in middle-late stages of the SOS response. Finally, these are disassembled, RecA storage structures are reformed and cells division resumes in the late stages of the SOS response.

DOI: https://doi.org/10.7554/eLife.42761.020

Heilemann lab. The PCR product was digested with *Not*I/*Bsr*GI and cloned in pBAD-*YPet-mcI* also digested with *Not*I/*Bsr*GI. The resulting construct was sequence verified prior to imaging and found to carry the correct PAmCherry1 sequence.

## Cloning of pETMCSI-*mcI*, pETMCSI-*YPet-mcI* and pND706-*PAmCherry-mcI* for protein purification

Expression vectors carrying the *mcI*, *YPet-mcI* and *PAmCherry-mcI* genes with no additional purification tags were created by sub cloning sequence verified geneblocks into pETMCSI (*Nde*I/*Eco*RI) or pND706 (*Nde*I/*Eco*RI). Clones were verified by PCR and sequencing. The pETMCSI-*mcI* and pETMCSI-*YPet-mcI* were found to be correct. *PAmCherry-mcI* could not be cloned into the pETMCSI expression vector despite several attempts. We therefore cloned it into the pND706 expression vector that carries the gene under the λ-promoter. The resulting PAmCherry-mCI construct was found to carry an R202H mutation in the PAmCherry protein. This construct was purified and found to be fluorescent *in vitro*.

## Creation of RBS mutants of pJM1071 and RecA overexpressor plasmids

pJM1071 is a low copy plasmid that carries a pSC101 origin of replication, spectinomycin marker and the constitutive *recA* promoter (*Churchward et al., 1984*). Plasmid pConst-*mKate2* was created by amplifying *mKate2* with primers (see supplemental table 3 in *Supplementary file 1*) and sub-cloning into the *Nde*I/*Xba*I sites of pJM1071. pG353C-*mKate2* was created by employing the Quick-Change protocol to introduce point mutations in the ribosomal binding site of the *recA* gene (AGGAGTAA) in the parent vector so that the mutant RBS encodes AGGACTAA instead. Introduction of this mutation reduces the expression of downstream gene products by two fold (see *Figure 6—figure supplement 1A*).

RecA overexpression plasmids were created by cloning the *recA* gene (obtained as a sequence verified geneblock) between the *NdeI/XbaI* sites of pJM1071 or pG353C-*mKate2*. Constructs were verified by sequencing prior to imaging in live-cells to ensure that the promoter region and the coding sequences were unchanged.

## Genetic recombination

MG1655 *recA-gfp* strain was a gift from the Weibel lab. MG1655 *dnaQ-YPet* was constructed previously (*Robinson et al., 2015*). RW1572 was created by P1 transduction of *lexA3*(Ind⁻) *malB*::Tn*9* from DE407 into MG1655 *dnaQ-YPet* by selecting for chloramphenicol resistance and then screening for *lexA3*(Ind⁻) associated UV sensitivity. MG1655 Δ*uvrD* was created by replacing the *uvrD* gene with a *FRT-Kan-FRT* marker using standard λ RED recombination (*Datsenko and Wanner, 2000*).

To construct the strain EAW767, EAW428 strain, which carries *recA-gfp* under the wild-type *recA* promoter, was first constructed using a modification of the procedure by Datsenko and Wanner (*Datsenko and Wanner, 2000*). A plasmid, pEAW944, was used as the template in a PCR to generate a product consisting of the C-terminus of *recA* fused to GFP, followed by the *recX* gene, then the pJFS42 mutant FRT-KanR-wt FRT cassette, and finally by 207 bp of the chromosome just downstream of the stop codon of the *recX* gene. The PCR product was electroporated into MG1655/pKD46 and a KanR colony was sequenced to confirm the presence of the *recA-GFP* fusion. The KanR was removed from the mutant *FRT-KanR-wt FRT* cassette, leaving a mutant FRT scar. The resulting EAW428 KanS strain was P1 transduced to Δ*dinI* with P1 grown on EAW736, which is a Δ*dinI* strain made by the method of *Datsenko and Wanner, 2000*. The resulting strain, EAW767 was sequenced to confirm the presence of both the *recA-GFP* fusion, and the Δ*dinI* allele.

## Expression and purification of mCI and fluorescently tagged mCI

See section on 'Construction of vectors' for details on cloning of over-expression vectors for protein purification. Overproduction of mCI, YPet–CI, and PAmCherry–mCI was achieved in *E. coli* strain BL21(DE3) *recA* containing plasmid derivatives of vector pND706 (*Love et al., 1996*) or pETMCSI (*Neylon et al., 2000*). Briefly, mCI and YPet–mCI were grown at 37°C in LB medium supplemented with thymine (25 mg/L) and ampicillin (100 µg/L). Upon growth to $A_{600}$ = 0.6, 1 mM isopropyl-β-D-thiogalactoside (IPTG) was added and cultures were shaken for a further 3 hr, then chilled on ice. Cells were harvested by centrifugation (16,900 ×*g*; 8 min), frozen in liquid N₂ and stored at –80°C. PAmCherry–mCI was grown at 30°C in LB medium supplemented with thymine (25 µg/L) and ampicillin (100 mg/L). Upon growth to $A_{600}$ = 0.7, to induce overproduction of PAmCherry–mCI the temperature was rapidly increased to 42°C. Cultures were shaken for a further 3 hr, and then chilled on ice. Cells were harvested by centrifugation (16,900 ×*g*; 8 min), frozen in liquid N₂ and stored at –80°C.

After thawing, cells (8 g) containing mCI were resuspended in 125 mL of lysis buffer (50 mM Tris/HCl, pH 7.6, 1 mM EDTA, 2 mM dithiothreitol, 20 mM spermidine, 100 mM NaCl). This solution was supplemented with three protease inhibitor cocktail tablets (Complete C, Roche) and 0.7 mM phenylmethanesulfonyl fluoride to inhibit proteolysis of mCI. The cells were then lysed by being passed twice through a French press (12,000 psi). The lysate was clarified by centrifugation (35,000 ×*g*; 30 min) to yield the soluble Fraction I. Proteins that were precipitated from Fraction I by addition of solid ammonium sulfate (0.4 g/mL) and stirring for 60 min were collected by centrifugation (35,000 ×*g*; 30 min) and dissolved in buffer A (50 mM Tris/HCl, pH 7.6, 1 mM EDTA, 2 mM dithiothreitol, 5 % (*v/v*) glycerol) supplemented with 130 mM NaCl. The solution was dialysed against 2 L of the same buffer, to yield Fraction II.

Fraction II was applied at 1.5 mL/min onto a column (2.5 × 10 cm) of DEAE-650M resin that had been equilibrated in the same buffer. Fractions containing proteins that did not bind to the column were pooled and dialyzed against one change of 2 L of buffer A. The dialysate (Fraction III) was loaded at a flow rate of 2 mL/min onto a column (2.5 × 10 cm) of the same resin, now equilibrated in buffer A. After the column had been washed with 155 mL of the same buffer, proteins were eluted using a linear gradient (600 mL) of 0–300 mM NaCl in buffer A. mCI eluted in a peak at about 30 mM NaCl. Fractions containing mCI were pooled and dialyzed against two changes of 2 L of buffer B (30 mM Tris/HCl, pH 7.6, 1 mM EDTA, 2 mM dithiothreitol, 5% (*v/v*) glycerol). The dialysate (Fraction IV) was applied at 1 mL/min onto a column of MonoQ 10/100 GL equilibrated in buffer B. After

the column had been washed with 30 mL of the same buffer, proteins were eluted using a linear gradient (150 mL) of 0–250 mM NaCl in buffer B. mCI eluted in a peak at 140 mM NaCl. Fractions containing mCI were pooled and dialyzed in 2 L of storage buffer (50 mM Tris/HCl, pH 7.6, 1 mM EDTA, 3 mM dithiothreitol, 100 mM NaCl, and 20% (v/v) glycerol) to yield Fraction V. Aliquots were frozen in liquid $N_2$ and stored at – 80°C. YPet–mCI and PAmCherry–mCI were purified according to the procedure used for mCI. The molecular mass of purified mCI, YPet–CI, and PAmCherry–mCI were verified by electro-spray ionisation mass spectrometry.

## Surface Plasmon Resonance (SPR) experiments

SPR experiments were performed on BIAcore T200 instrument (GE Healthcare) using streptavidin (SA) coated sensor chip to study the binding kinetics of mCI, YPet-mCI and PAmCherry-mCI to RecA filaments assembled on ssDNA. All experiments were carried out at 20°C with a flow rate of 5 µL/min, unless specified otherwise. SA chip was activated with three sequential 1 min injections of 1 M NaCl, 50 mM NaOH, then stabilized by 1 min treatment with 1 M $MgCl_2$. A single-stranded biotinylated 71-mer poly-dT oligonucleotide bio-$(dT)_{71}$ was diluted to 5 nM in SPR buffer (30 mM Tris/HCl, pH 7.6, 50 mM NaCl, 5 mM $MgCl_2$, 0.005% (v/v) surfactant P20, 0.5 mM dithiothreitol) and immobilised (137 RU) onto the flow cell of the SA chip by flowing it at 10 µL/min for 300 s. RecA filaments were then assembled on bio-$(dT)_{71}$ (~2240 RU in each case) by injecting a solution of 1 µM RecA in SPR$^{RecA}$ buffer (20 mM Tris/HCl, pH 8.0, 10 mM KCl, 10 mM $MgCl_2$, 0.005% surfactant P20 and 0.5 mM dithiothreitol) supplemented with 1 mM ATP at 10 µL/min for 300 s.

Immediately following the RecA-ssDNA filament assembly in each case, binding studies were performed by injecting 2 µM mCI/YPet-mCI/PAmCherry-mCI in SPR$^{RecA}$ buffer supplemented with 1 mM ATPγS for 400 s (including blank injection of just buffer), followed by co-injection of the same buffer with ATPγS for 1000 s to monitor dissociation of bound proteins from RecA (*Figure 2—figure supplement 1C*). Regeneration of the surface in preparation for the next round of RecA filament assembly was performed by treatment with 1 M $MgCl_2$ for 1 min. An unmodified flow cell (without immobilised oligonucleotide) was used as a reference surface and subtracted from all signals. Resulting SPR signals in all four cases were further normalized to the signal of RecA assembled on bio-$(dT)_{71}$ prior to injection of each binding partner and then subtracted from the blank sensorgram for drift correction due to slow disassembly of RecA filaments in the presence of ATPγS in SPR$^{RecA}$ buffer. For purposes of presentation and comparison of dissociation rates, sensorgrams were scaled to the highest signal of YPet-mCI bound over the RecA-$(dT)_{71}$ filament, and these are presented in *Figure 2B*. RecA filaments were assembled in SPR buffer containing ATP because the assembly of proper RecA filaments on ssDNA is a dynamic process involving ATP hydrolysis. However, once assembled, binding studies of mCI and analogues were performed in the presence of ATPγS, which slows down RecA filament disassembly.

## Single-molecule FRET imaging

### Preparation of FRET substrate

Biotinylated DNA substrates for FRET were designed as by hybridizing the oligos (IDT, USA):

AS18_Cy3: TGG CGA CGG CAG CGA GGC/3Cy3Sp/

Cy5_S18dT40_bio:/5Cy5/TT TTT TTT TTT TTT TTT TTT TTT TTT TTT TTT TTT TTT TTG CCT CGC TGC CGT CGC CA/3Bio/

Bio-ds18-$(dT)_{40}$ FRET substrate was assembled by hybridizing AS18_Cy3 and Cy5_S18dT40_bio in hybridization buffer (10 mM Tris/HCl, pH 7.5, 50 mM NaCl, 5 mM $MgCl_2$ and 0.5 mM EDTA) at a final concentration of 100 µM, by heating at 95°C for 10 min followed by annealing over a course of three hours.

### Microscope setup for FRET imaging

A home built objective-type TIRF microscope based on an Olympus IX-71 model was used to record single molecule movies. Sapphire, green (532 nm) laser was used to excite donor molecules by focusing on to 100X oil immersed objective. FRET was measured by excitation with a 532 nm laser and the emissions at 555 and 647 nm were collected using a band-pass filter at 555 nm and a long-pass filter at 650 nm. Scattered light was removed by using 560 nm long pass filter. Cy3 and Cy5 signals were separated by 638 nm dichroic using photometrics dual view (DV-2) and both signals were

focused onto CCD camera (Hamamatsu C9 100–13), simultaneously. Data were collected at 5–10 frames/s.

## Sample preparation for FRET experiments

Quartz coverslips were treated with 100% ethanol and 1 mM KOH. Then, aminosilanization of coverslips was carried out in a 1% (v/v) (3-Aminopropyl)triethoxy silane (Alfa Aesar, A10668, UK) solution in acetone. PEGylation was carried out by incubating mixture of biotinPEG-SVA and mPEG-SVA (Laysan Bio, AL) in the ratio of 1:10 prepared in 0.1 M NaHCO$_3$ solution on the top of silanized coverslip for 3–4 hr. Finally, PEGylated coverslips were stored under dry nitrogen gas at −20°C.

## Single-molecule FRET experiments

Immuno-pure streptavidin solution was prepared in RecA buffer (20 mM Tris/HCl, pH 8.0, 10 mM KCl and 5 mM MgCl$_2$) and spread on the top of dry PEGylated coverslip followed by a 10 min incubation. Sample flow chambers were created by sandwiching polydimethylsiloxane (PDMS) on the top of the streptavidin coated coverslip. Then, blocking buffer (20 mM Tris/HCl, pH 8.0, 10 mM KCl, 5 mM MgCl$_2$, 0.2 mg/mL BSA, 0.25% (v/v) Tween 20) was injected into the channel in order to reduce non-specific binding of proteins on the surface followed by 10–15 min incubation. A 50 pM solution of bio-ds18-(dT)$_{40}$ substrate was prepared in RecA buffer and injected into the flow chamber using a syringe pump (ProSense B.V.) followed by incubation for 10 min. Unbound DNA was washed off in RecA buffer, and RecA-ATP filaments were formed on surface immobilized DNA substrate by injecting RecA buffer supplemented with 1 mM ATP and 1 μM RecA. Movies were recorded at room temperature (20 ± 1°C) for 2–3 min in oxygen-scavenging system (OSS) consisting of protocatechuic acid (PCA, 2.5 mM) and protocatechuate-3,4-dioxigenase (PCD, 50 nM) to reduce photo-bleaching of the fluorophores and 2 mM Trolox to reduce photo-blinking of dyes. For mCI titration experiments, once RecA filaments were formed, different concentrations of mCI proteins prepared with 1 μM ATP and 1 μM RecA and OSS solution in RecA buffer were injected using a syringe pump. The concentration of RecA and ATP was maintained while recording movies.

## Data analysis

Single-molecule intensity time trajectories were generated in IDL and these trajectories were analyzed in MATLAB using home written scripts. Approximate FRET value is measured as the ratio of acceptor intensity to the sum of the donor and acceptor intensities after correcting cross talk between donor and acceptor channels. To measure kinetics of RecA filament assembly and disassembly, a cut off FRET value was chosen (0.3 FRET), so that states with FRET values above the threshold represented unbound substrates, and those below the threshold represented the bound state. *Figure 2* was prepared using home written script (*Shen et al., 2012*).

## Measurement of the influence of mCI on RecA* ATPase activity

Reactions were carried out at 37°C and contained 25 mM Tris/HCl, pH 7.5, 10 mM MgCl$_2$, 3 mM potassium glutamate, 5% (v/v) glycerol, 1 mM dithiothreitol, a coupling system (3 mM phosphoenolpyruvate, 10 U/mL pyruvate kinase, 2 mM NADH, 10 U/mL lactate dehydrogenase), 5 μM M13mp18 circular ssDNA, 3 μM RecA protein, 3 mM ATP, 0.5 mM SSB, and the indicated concentration of mCI protein. Reactions were started with ATP and SSB, and were continuously monitored at a wavelength of 340 nm.

## Measurement of the influence of mCI on RecA* mediated LexA cleavage

Reactions were carried out at 37°C and contained 40 mM Tris/HCl, pH 8.0, 10 mM MgCl$_2$, 30 mM NaCl, 2 mM dithiothreitol, 3 mM ATP, 5 μM M13mp18 circular ssDNA, 3.5 μM RecA protein, 8 μM LexA protein, and the indicated concentration of mCI protein in a total volume of 90 μL. Aliquots (20 μL) were taken at the times indicated, and run on a 12% PAGE gel.

## Measurement of the influence of mCI on strand exchange activity of RecA*

Reactions were carried out at 37°C and contained 25 mM Tris/HCl (pH 7.5), 10 mM $MgCl_2$, 3 mM potassium glutamate, 5% (v/v) glycerol, 1 mM dithiothreitol, an ATP regeneration system (2.5 mM phosphoenolpyruvate and 10 U/mL pyruvate kinase), 10 µM M13mp18 circular ssDNA, 6.5 µM RecA protein, and 10 µM mCI protein. After 10 min pre-incubation at 37°C, ATP and SSB were added at 3 mM and 1 µM respectively to allow RecA filament formation. Reactions were started 10 min later with addition of 20 µM linear dsDNA (also M13mp18). Reaction volume was 60 µL and 12 µL aliquots were taken at the times indicated. Reactions were stopped (ficoll/SDS/proteinaseK) and run on a 0.8% agarose gel.

## UV-survival assay

UV survival assay was performed by growing indicated strains in LB medium at 30°C in the indicated amount of L-arabinose in a 500 µL culture. Cell cultures exhibiting an absorbance measured at 600 nm excitation ($A_{600}$) between 0.4 and 1 OD were pelleted. The cell pellet was washed 2X in 100 mM $MgSO_4$ and resuspended in 600 µL of 100 mM $MgSO_4$ and $A_{600}$ of the cell suspension was measured. Cells were diluted in 100 mM $MgSO_4$ to obtain an OD 0.25 suspension. Cells were then irradiated by placing a 10 µL drop on a quartz piece and exposing to 10, 20, 40 or 80 $Jm^{-2}$ of UV-254 from a Herolab UV-8-S/L UV lamp (Wiesloch, Germany). UV fluence was measured prior to each experiment, with a UVX radiometer equipped with a UVX-25 shortwave sensor (UVP, CA, USA). Cells were serially diluted (1:10) 4X, and 20 µL of the fourth dilution were plated in duplicate on LB plates supplemented with ampicillin (100 µg/mL) for each condition. After incubation at 30°C for 16 hr, colonies were counted. The number of colonies counted for each UV dose was normalized to the no UV condition for each strain, to obtain the fractional survival. Experiments were repeated two times (two technical replicates, two independent experiments) and three times (two technical replicates, three independent experiments) for HG116 and HG267 respectively. These are plotted in *Figure 4— figure supplement 1A*. Error bars represent standard error of the mean.

## Live-cell imaging

Live cell imaging was performed on a custom built TIRF microscope equipped with 405 (OBIS, Coherent, CA), 488 (Sapphire, Coherent, CA), 514 (Sapphire, Coherent, CA), 568 (Sapphire, Coherent, CA) and 647 (OBIS, Coherent, CA) nm lasers. Cellular imaging was performed using near-TIRF conditions enabling illumination of bacterial cells immobilized on the surface using an inverted fluorescence microscope (Nikon Eclipse-Ti) equipped with a 1.49 NA 100X objective and a 512 × 512 $px^2$ Photometrics Evolve CCD camera (Photometrics, AZ). NIS-Elements equipped with JOBS module was used to operate the microscope (Nikon, Japan).

488 nm laser light was directed through a 405/488/561/647 (Chroma, Vermont) dichroic prior to sample excitation and collected using a 488 LP filter (Chroma, Vermont). The 514 nm laser light was directed through a 405/514/568 dichroic and ET535/30 m emission filter (Chroma, Vermont). The photo-activatable channel was equipped with a 405 nm OBIS laser (200 mW max. output, Coherent, CA) and a 568 nm Sapphire LP laser (200 mW max. output, Coherent, CA). The emission filter is ET590LP (Chroma, Vermont).

Custom flow-cells for live-cell imaging were constructed by gluing an (3-Aminopropyl)triethoxy silane (Alfa Aesar, A10668, UK) treated coverslip (Marienfeld, Deckglaser, 24 × 50 mm No. 1.5, Germany) to a quartz piece (Proscitech, Australia) using double-sided sticky tape (970XL ½ X 36yd , 3M) to create a channel, and sealed with epoxy. Quartz pieces were designed to provide an inlet and outlet tubing (PE-60, Instech Labs).

Prior to imaging, cells were revived from a −80°C freezer stock by resuspension in LB liquid media overnight (0.5 mL) and shaking at 1000 rpm at 30°C on an Eppendorf thermomixer C (Eppendorf, Australia). On the following day, cells were diluted in EZ rich defined medium supplemented with 0.2% (v/v) glycerol as the carbon source (EZ-glycerol). When necessary, cells were induced with the indicated amount of L-arabinose and allowed to grow over night at 30°C. On day 3, cells were diluted 1:200 or 1:1000 in 500 µL of EZ-glycerol and allowed to grow for 4–6 hr or until the cells reached OD 0.2–0.4. Cells were then flowed into the flow cell and allowed to settle on to the surface of the cover-slip. Finally, the cell suspension was switched with fresh growth medium before starting

the experiment. Imaging was performed at 30°C on a heated stage coupled to a heated objective (OKO Labs).

In situ UV-irradiation was provided by means of a UV pen ray light source (UVPA90-0012-01, 11SC-1, 254 nm, Upland CA, USA) with a G-275 filter allowing transmission of 254 nm UV. UV flux was measured using a UVX radiometer (UVP, Australia) and appropriate exposure times were determined to provide the required dosage. UV-irradiation was performed by shuttering the UV-lamp and allowing exposure according to the time calculated for each dose.

## Measurement of cellular fluorescence intensity

To measure the fluorescence intensity in cells, time-lapse experiments were performed in flow-cells and cells were imaged every 5 min for 3 hr after UV-irradiation at 12 or 30 distinct (x, y) positions in the flow-cell. At each location at each time point, one bright-field image and one image in the GFP channel (Sapphire laser 488 nm excitation, Dichroic: 405/488/561/647, Emission filter: 488 LP; 100 ms exposure EM gain: off, 12 W/cm$^2$).

Cell outlines were identified in bright-field images using custom software in Fiji (*Schindelin et al., 2012*) and MATLAB (MathWorks). Automatic cell detection was performed as follows:

1. Brightfield images were first filtered using a Bandpass filter: run('Bandpass Filter...', 'filter_-large = 2 filter_small = 0 suppress = None tolerance = 5 autoscale saturate');
2. Following this, edges were detected using FeatureJ (http://imagescience.org/meijering/software/featurej/ ) as follows: run('FeatureJ Edges', 'compute smoothing = 2 lower=[] higher=[]'); run('8-bit');
3. Finally, ridge detection (*Steger, 1998*) was performed: run('Ridge Detection', 'line_width='+linewidth + " high_contrast='+high_contrast + " low_contrast='+low_contrast + " darkline extend_line       show_junction_points       displayresults       add_to_manager method_for_overlap_resolution = SLOPE sigma='+sigma + " lower_threshold='+low_threshold + " upper_threshold='+upper_threshold);

Typical values for these parameters were:

linewidth = 2;

high_contrast = 230;

low_contrast = 87;

sigma = 0.8;

low_threshold = 1.7;

upper_threshold = 5;

Cell outlines were then used as ROIs in Fiji to measure mean cellular fluorescence intensity of cells in the GFP channel averaged over the area of the cell, at each time point.

In *Figure 1*, traces represent the fold increase ($FI$) in the average fluorescence detected in cells at the indicated time point ($F_i$) normalized to the average fluorescence detected in the absence of damage ($F_0$). For each time point $i$, the average fluorescence ($F_i$) was calculated as the product given by area of cell ($A_{k,i}$) x mean fluorescence ($I_{k,i}$), averaged over all the cells ($K$) detected for time point $i$, $K_i$.

$$F_i = \frac{\sum_{k=1}^{K} A_{k,i} I_{k,i}}{K_i}$$

$$FI_i = \frac{F_i}{F_0}$$

Error bars ($\Delta F_i$) were calculated as standard deviation ($\sigma_{F_i}$) of the fluorescence of cells as follows, where: $\hat{A}$ represents the mean area, and $\hat{I}$ represents the mean intensity at that time point $i$.

$$\sigma_{F_i} = \sqrt{\frac{\sum_{k=1}^{K} \left( A_{k,i} I_{k,i} - \left( \hat{A}_i \hat{I}_i \right) \right)^2}{(K_i - 1)}}$$

$$\Delta F_i = \frac{\sigma_{F_i}}{\sqrt{K_i}}$$

$$\Delta FI_i = \Delta F_i + \Delta F_0$$

In Figure 3, traces represent the average fluorescence detected in cells at the indicated time point. For each time point $i$, the average fluorescence ($F_i$) was calculated as the product given by area of cell ($A_{k,i}$) x mean fluorescence ($I_{k,i}$), averaged over all the cells ($K$) detected for time point $i$, $K_i$.

$$F_i = \frac{\sum_{k=1}^{K} A_{k,i} I_{k,i}}{K_i}$$

Error bars were calculated as standard deviation of the fluorescence of cells as follows, where: $\hat{A}$ represents the mean area, and $\hat{I}$ represents the mean intensity at that time point $i$.

$$\sigma_{F_i} = \sqrt{\frac{\sum_{k=1}^{K} \left( A_{k,i} I_{k,i} - \left( \hat{A}_i \hat{I}_i \right) \right)^2}{(K_i - 1)}}$$

$$\Delta F_i = \frac{\sigma_{F_i}}{\sqrt{K_i}}$$

## Measurement of co-localization of PAmCherry-mCI and ε-YPet in live cells

Experiments to measure co-localization of PAmCherry-mCI and ε-YPet were performed in a flow cell with cells growing at 30°C with a continuous supply of aerated growth medium (EZ Glycerol). To identify whether PAmCherry-mCI co-localizes with ε-YPet in individual cells, a three-phase acquisition protocol was employed: first, a 100 ms snap-shot of cells was taken in the bright-field channel to yield cell outlines. In the second phase, we used a PALM acquisition protocol to image the PAm-Cherry-mCI (*Figure 4—figure supplement 1B*). PAmCherry was detected by observing cells under simultaneous illumination with the activation laser 405 (1–5 W/cm$^2$) and 568 nm readout laser (540 W/cm$^2$) for 200 frames (100 ms each). In this time frame, activated PAmCherry-mCI was either diffusive in the cytosol, or formed bright foci. To minimize laser damage from imaging, this imaging protocol was executed exactly once on any given set of cells at any time point, and a new group of previously unexposed cells was imaged for every subsequent time point (*Figure 4—figure supplement 1B*). ε-YPet was detected using a 514 nm laser (~2200 W/cm$^2$) in a 100 ms exposure.

Co-localization analysis was performed in Fiji. First cell outlines were drawn in MicrobeTracker (*Sliusarenko et al., 2011*) on the brightfield images. Cell outlines were imported into Fiji and analyzed using custom codes that yielded peak positions (*Caldas et al., 2015*). Co-localization of PAm-Cherry-mCI with replisome foci was measured by first creating a single maximum intensity projection of the 568 acquisition data that revealed the intra-cellular sites of PAmCherry-mCI foci and interrogating whether these foci co-localized with replisome foci in the 514 channel within a radius of 200 nm. Since arabinose expression is highly heterogeneous at low concentrations of L-arabinose, only those cells that exhibited at least one focus in both channels were used for analyses.

## Measurement of chance colocalization

Treating the probabilities of observation of mCI foci and replisomes as independent events, chance co-localization at each time point was calculated as:

$$p\left(mcl \, and \, replisome\right) = p\left(mcl\right) p\left(replisome\right)$$

$$P(mcl) = \frac{\#mcl_{cell}^{foci} \times total \, \#cells \, \times area \, of \, each \, focus}{total \, area \, of \, cell}$$

$$P(replisomes) = \frac{\#replisome\frac{foci}{cell} \times total\ \#cells\ \times area\ of\ each\ focus}{total\ area\ of\ cell}$$

For the purpose of this calculation, the area of each focus was assumed to be 0.28462 µm$^2$ (25 px$^2$). Total numbers of cells are provided in *Figure 4—figure supplement 1D*. Cell areas are provided in *Figure 4—figure supplement 1F* and chance colocalizations are provided in *Figure 4—figure supplement 1G* (note the scale on Y-axis is 0–0.1 to emphasize the traces).

## Construction of over-expresser strains of RecA

First, we cloned the wild-type *recA* gene into a low copy plasmid such that RecA is expressed from the constitutive *recAo281 operator,* that carries an AT to GC transition at position −17 with respect to the transcription start site ('pConst-*recA*'; strain# HG411) (*Uhlin et al., 1982*; *Volkert et al., 1976*). pConst-*recA* has a pSC101 origin of replication and its copy number is expected to be approximately six copies per cell (*Cabello et al., 1976*; *Hasunuma and Sekiguchi, 1977*; *Peterson and Phillips, 2008*). With the objective of reducing the translation efficiency of the transcript expressed from the *recAo281* promoter, we created a second plasmid (pG353C-*recA*) derived from pConst-*recA* that carries a G to C transversion in the ribosome binding site (RBS) at position −6 (with respect to the translation start site). Since mutations in the RBS strongly influence the complementarity to the 16S rRNA and consequently influence the efficiency of translation, we anticipated that this G to C transversion would reduce the expression level from the strong *recA* RBS (*Johnson et al., 1991*; *Shine and Dalgarno, 1974*; *Shine and Dalgarno, 1975*). To validate the strength of the constitutive *recAo281* promoter in the wild-type or mutant RBS background, we additionally derived two vectors from pConst-*recA* and pG353C-*recA* that express red fluorescent protein mKate2 instead of RecA. These vectors pConst-*mKate2* (n = 92 cells) and pG353C-*mKate2* (n = 92 cells) were then imaged in wild-type cells, and the cytosolic fluorescence intensity was measured in the absence of DNA damage. We found that introduction of the single point mutation in the RBS reduced the constitutive expression of mKate2 by half (see *Figure 6—figure supplement 1A*). Based on these results, we expect the expression of RecA from the pG353C-*recA* plasmid to be half of that from pConst-*recA* in cells.

## Measurement of Feret diameters of RecA-GFP storage structures

Dimensions of storage structures were analysed at a threshold of 4000 au. For reference the background intensity of the slides is 479 ± 28 au (n = 32 cells), intensity of the cytosolic fluorescence of RecA-GFP was 1576 ± 402 au (n = 32 cells) and intensity of RecA in storage structures was 8615 ± 3101 (n = 32 cells). At this threshold, the Feret diameter along the longest dimension was measured for MG1655 *recA-gfp* (n = 528), *recA-gfp*/pG353C-*recA* (n = 137) and *recA-gfp*/pConst-*recA* (n = 399) storage structures. The measured Feret diameter exhibited a clear dependence on the amount of excess untagged wild-type RecA (see *Figure 6*).

## Measurement of dynamics of storage structure dissolution and sequestration

Storage structures were found to dynamically dissolve and re-appear upon UV-irradiation in time-lapse experiments with frames taken every 5 min. We chose to analyse the dynamics of RecA redistribution in *recA-gfp*/pG353C-*recA* cells since these cells exhibited a single large rod-shaped structure that encapsulated a majority of the RecA-GFP fluorescence (*Figure 6A*). All the cells imaged in these experiments exhibited storage structures at some point in the cell-cycle. To quantify the timescales of storage structure dissolution, we measured the time taken for the intensity of the storage structure to drop such that it was indistinguishable from the cellular background in *recA-gfp*/pG353C-*recA* cells. Conversely, reassembly dynamics of storage structures were characterized by calculating the time taken for the appearance of a storage structure after UV-irradiation at t = 0.

A total of 929 storage structures were analysed. 485 storage structures were present at the start of the experiment, of which 213 did not dissolve. 56% of the storage structures present at the start of the experiment (272 out of 485) were found to dissolve and the dynamics of the dissolution are presented in *Figure 6D*. 444 storage structures were found to appear during the course of the

experiment. Data were compiled from five independent time-lapse experiments and represent greater than 300 cells in total that are present for the duration of the experiment.

Time of onset of RecA-bundles was measured in MG1655 RecA-GFP cells. For this measurement, cells were observed in time-lapse experiments with frames acquired every 5 min. During RecA bundle formation, punctate foci become linear structures that form mature RecA-bundles over several tens of minutes (see *Figure 5—video 1*). The time taken for the first appearance of a linear morphology was measured for 108 RecA-bundles. Cumulative frequency of the appearance of RecA-bundles is plotted in *Figure 6H*. Error bars represent standard deviation of the mean of the bootstrap distribution obtained by bootstrapping 80% of the data set, 1000X to obtain the mean.

## Electron microscopy to detect storage structures in cells carrying untagged RecA

E.*E. coli* strains (HH020, HG001, HG411 and HG465) were grown overnight at 30°C in LB medium containing appropriate antibiotics (where necessary). The overnight culture was diluted 1:100 into 5 mL fresh LB media and grown at 30°C for approximately 4.5 hr until they reached an $OD_{600}$ ~0.4–0.5. 1 mL of the culture was transferred to a microfuge tube and centrifuged at ~14,000 x g for 5 min. The supernatant was decanted and the pellet resuspended in an equal volume of PBS containing 0.5% glutaraldehyde. The suspension was kept on ice for 25 min before centrifugation and washed 2X with an equal volume of PBS. Cells were embedded in resin and sectioned by Jan Endlich (JFE Enterprises, Beltsville, MD) under a custom service contract. This process entailed lightly centrifuging the cells into a pellet, which were dehydrated in a series of ethanols, (30% EtOH, 50% EtOH and 70% EtOH). Cells were then infiltrated 1:1 with LR White to 70% EtOH, 1:2 LR White to 70% EtOH, 1:3 LR White to 70% EtOH and 3 changes of pure resin. The infiltrated samples were then transferred in gelatin capsules with fresh LR White and then cured for 48 hr at 50°C. Blocks were removed and gelatin capsule sections were cut approximately 60–80 nm thin. Sectioned EM grids were returned to us and were placed on 50 µL droplets of sterile PBS containing 0.5% BSA and 0.1% gelatin (PBSBG) for 3 hr at room temperature to reduce any non-specific binding of the RecA antibodies. After this time, the grids were placed on a 50 µL droplet of a (1:25, 1:100, 1:500) dilution of affinity purified RecA polyclonal rabbit antisera diluted in PBSBG and incubated overnight at room temperature (~22°C). Grids were washed 3X by placing on sequential drops of 50 µL PBSG for 10 min each. Grids were then placed on a 50 µL droplet of a 1:40 dilution of goat-anti-rabbit antibody (in PBSBG) conjugated to 30 nm gold particles (Abcam, cat # ab119178) and incubated overnight at room temperature. The grids were washed 3X with PBSG for 10 min each and dried by blotting on filter paper. Grids were then returned to Jan Endlich (JFE enterprises, Beltsville, MD), who stained them for 5 min with 2% aqueous uranyl acetate and then examined and imaged the cells with a Zeiss EM 10C transmission electron microscope.

## DinI experiments

Since storage structure formation is sensitive to the expression level of RecA, we created strain EAW428 that expresses RecA-GFP from the wild-type *recA* promoter (henceforth referred to as $P_{Wt}$-*recA-gfp*) on the chromosome. To identify whether DinI expression influenced RecA structures in cells, we created strain EAW767 ($P_{wt}$-*recA-gfp* Δ*dinI*). Comparison of fluorescence images of EAW428 and EAW767 grown under identical conditions revealed that EAW767 cells exhibited fewer RecA-GFP structures in the absence of DNA damage. On average, 27% of EAW767 ($P_{wt}$-*recA-gfp* Δ*dinI*) cells exhibited structures ($n_{cells}$ = 702 cells), compared to 43% of EAW428 ($P_{wt}$-*recA-gfp*) cells ($n_{cells}$ = 855). Structures were identified using a threshold of 1200 and the Feret diameters of these fluorescent features measured. Comparison of the probability distribution functions of the Feret diameters revealed that Δ*dinI* cells did not produce RecA structures of Feret diameters that were statistically significantly different from those observed in RecA structures in *dinI*[+] cells in the *recA-gfp* background.

## Acknowledgements

We thank Douglas Weibel for the MG1655 RecA-GFP strain, Alon lab for the SOS-reporter plasmids and Nick Dixon for pETMCSI and pND706 plasmids. We thank Amy McGrath and Celine Kelso for technical assistance with ESI-MS analyses of purified proteins. RW was supported by the NICHD/NIH

Intramural Research Program. MMC is supported by NIH Grant U01 GM32335 from the National Institute of General Medical Sciences. AMVO acknowledges support by the Australian Research Council (DP180100858 and FL140100027).

## Additional information

### Funding

| Funder | Grant reference number | Author |
|---|---|---|
| National Institutes of Health | Intramural Research Program | Roger Woodgate |
| National Institutes of Health | U01 GM32335 | Michael M Cox |
| Australian Research Council | DP180100858 | Antoine M van Oijen |
| Australian Research Council | FL140100027 | Antoine M van Oijen |

The funders had no role in study design, data collection and interpretation, or the decision to submit the work for publication.

### Author contributions

Harshad Ghodke, Conceptualization, Data curation, Software, Formal analysis, Investigation, Methodology, Writing—original draft, Writing—review and editing; Bishnu P Paudel, Data curation, Software, Formal analysis, Investigation, Writing—review and editing; Jacob S Lewis, Investigation, Writing—review and editing; Slobodan Jergic, Data curation, Formal analysis, Investigation, Writing—review and editing; Kamya Gopal, Zachary J Romero, Investigation; Elizabeth A Wood, Resources; Roger Woodgate, Resources, Data curation, Investigation, Writing—review and editing; Michael M Cox, Conceptualization, Funding acquisition, Writing—review and editing; Antoine M van Oijen, Conceptualization, Supervision, Funding acquisition, Writing—review and editing

### Author ORCIDs

Harshad Ghodke http://orcid.org/0000-0002-6628-876X
Bishnu P Paudel https://orcid.org/0000-0003-3518-3882
Jacob S Lewis https://orcid.org/0000-0002-9945-6133
Slobodan Jergic http://orcid.org/0000-0001-5252-0815
Kamya Gopal http://orcid.org/0000-0002-8305-8384
Roger Woodgate http://orcid.org/0000-0001-5581-4616
Michael M Cox https://orcid.org/0000-0003-3606-5722
Antoine M van Oijen http://orcid.org/0000-0002-1794-5161

### Decision letter and Author response

Decision letter https://doi.org/10.7554/eLife.42761.025
Author response https://doi.org/10.7554/eLife.42761.026

## Additional files

### Supplementary files

• Supplementary file 1. Supplemental Tables. Supplemental Table 1: Description of plasmids used in this study. Supplemental Table 2: List of strains used in this study. Supplemental Table 3: List and sequences of primers used in this study. Supplemental Table 4: Sequences of plasmids used in this study. Supplemental Table 5: Sequences of inserts used for cloning.
DOI: https://doi.org/10.7554/eLife.42761.021

• Source code 1. Custom Matlab code and Fiji scripts used for analysis of cell fluorescence in *Figures 1* and *3*, and custom Matlab code and Fiji scripts used for co-localization analysis presented in *Figure 4*.
DOI: https://doi.org/10.7554/eLife.42761.022

• Transparent reporting form
DOI: https://doi.org/10.7554/eLife.42761.023

**Data availability**

All data generated or analysed during this study are included in the manuscript and supporting files. Codes used for analysis are publicly available (in GitHub as described in previous publications). Scripts using these codes are also now provided in this submission as Source code 1.

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
