## [Decision Letter]

Thank you for submitting your article "Spatial and temporal organization of RecA in the *Escherichia coli* DNA-damage response" for consideration by *eLife*. Your article has been reviewed by three peer reviewers, and the evaluation has been overseen by a Reviewing Editor and John Kuriyan as the Senior Editor. The following individual involved in review of your submission has agreed to reveal their identity: Eric C Greene (Reviewer #2).

The reviewers have discussed the reviews with one another and the Reviewing Editor has drafted this decision to help you prepare a revised submission.

Summary:

Essential revisions:

1) It is hard to envision the structural identity of the RecA bundles – they seem too large to simply be RecA bound to ssDNA. Can the authors measure the cross-section size of the bundles (based upon existing data) and confirm whether or not they are consistent with the expected width for a single ssDNA-bound RecA filament? To what extent do the cross-sections of the bundles differ from the cross-sections of the storage structures? Similarly, can the authors report the length of the RecA bundles, and based upon these results, estimate the amount of DNA that is contained within the bundles? RecA forms a filament on the ssDNA, and then this filament may polymerize into the flanking dsDNA. Is there a simple biochemical/biophysical test that the authors can perform to see whether mCI will interact with RecA that is bound to dsDNA?

2) This is a very difficult paper to understand. It needs major re-writing to clarify what the authors want to say. In general, there is not enough description and discussion of the experiments to understand why the authors are doing it, what they are doing and what they find. In addition, the videos in the supplemental material were very hard to understand – what was being shown and emphasized. Need more explanation and diagraming on the image. Some additional annotation to the videos by specifically highlighting (in the video itself) events or structures that the reader should focus on.

3) It is difficult to assess the importance of the RecA-GFP punctate foci that are measured in Figure 6. Isn't it possible that, rather than being physiologically relevant "storage structures", these are simply accumulations of the inactive fusion protein or accumulations of overexpressed protein? This concern is somewhat addressed by the EM images, but these EM images seem inconclusive – the fluorescence images have much more information. The main conclusion drawn from EM is that upon overexpression of RecA (again, not physiological conditions), the RecA goes to the membrane.

4) The section on the co-localization of the replication forks and the RecA structures is difficult to understand. The authors grow the cells in rich media (EZ-glycerol?) and so there are several round of replication going on at once. There are several replication forks and several RecA structures in a small cell. What is the probability that the overlap is random? Can they randomize the positions of the cells and foci and show that the overlap is not random? Alternatively, can they slow down growth and reduce number of forks and complexity and then look for overlap?

---

## [Author Response]

Essential revisions:1) It is hard to envision the structural identity of the RecA bundles – they seem too large to simply be RecA bound to ssDNA. Can the authors measure the cross-section size of the bundles (based upon existing data) and confirm whether or not they are consistent with the expected width for a single ssDNA-bound RecA filament? To what extent do the cross-sections of the bundles differ from the cross-sections of the storage structures? Similarly, can the authors report the length of the RecA bundles, and based upon these results, estimate the amount of DNA that is contained within the bundles?

Our strategy of sub-stoichiometric labelling of RecA structures using the mCI probe cannot adequately answer this question. On this matter, we direct the reader to the work described in Lesterlin et al. (2014) where the authors have used 3D-SIM to resolve these structures with higher spatial resolution than is accessible to us in our assays. In this work, the RecA-GFP bundles were described as having a thick central body with cross-section of 160 ± 30 nm and extensions less than 120 nm and a length of 1.4 μm. The authors proposed that RecA bundles may contain RecA that is not bound to DNA.

We have now included a new panel in Figure 5 showing a montage of a RecA-GFP bundle in a cell responding to UV damage. Our observations of RecA-GFP bundles (Figure 5—video 1, Figure 5A) reveal that RecA bundles exhibit different cross-sections along the length of the bundles, suggesting that they are not simply single RecA* filaments. These observations are consistent with Rajendram et al. (2015) and Lesterlin et al. (2014).

RecA forms a filament on the ssDNA, and then this filament may polymerize into the flanking dsDNA. Is there a simple biochemical/biophysical test that the authors can perform to see whether mCI will interact with RecA that is bound to dsDNA?

We can comment with certainty that the mCI probe stains RecA* in vitro. We have attempted to assemble RecA on a 60-mer dsDNA substrate in the presence of ATPγS on a SPR chip in a fashion analogous to the experiments described in Figure 2. In these experiments, we failed to collect sufficient signal indicating the formation of the RecA-dsDNA filament limiting our ability to perform experiments to assay mCI binding. The main conclusion from this experiment was that wild-type RecA does not assemble stably on dsDNA even in the presence of ATPγS. These data have not been included in this manuscript to avoid confounding an already complex narrative.

Conceivably, under conditions where wild-type RecA can assemble on dsDNA, mCI might be able to bind such filaments. We cannot comment on whether such conditions are achievable in vivo.

We have now included this discussion in the second paragraph of the subsection “In vitro characterization of the binding of mCI to RecA-ssDNA filaments”.

2) This is a very difficult paper to understand. It needs major re-writing to clarify what the authors want to say. In general, there is not enough description and discussion of the experiments to understand why the authors are doing it, what they are doing and what they find. In addition, the videos in the supplemental material were very hard to understand – what was being shown and emphasized. Need more explanation and diagraming on the image. Some additional annotation to the videos by specifically highlighting (in the video itself) events or structures that the reader should focus on.

We agree with the reviewers on this matter, and thank them for providing an opportunity to streamline the writing. We have implemented the following changes:

1) Long sections formerly composed of multiple studies are now broken into single sections with headings;

2) The section describing the RecA-GFP strain has now been elaborated to use it to frame the work in its constituent studies on (a) RecA* intermediates and (b) RecA storage structures;

3) Each experimental section has now been reformatted such that it describes the motivation of the experiment, the experimental setup and results and discussion pertaining to the experiment;

4) Cell outlines and arrows have now been annotated on the figures and videos to indicate the phenomena under discussion.

Additional changes in the interest of clarity:

1) Note that we have replaced the probability distribution function plot in Figure 6B with the raw data.

2) Figure 7 has now been modified to encapsulate the time-line of the SOS response (missing earlier).

3) It is difficult to assess the importance of the RecA-GFP punctate foci that are measured in Figure 6. Isn't it possible that, rather than being physiologically relevant "storage structures", these are simply accumulations of the inactive fusion protein or accumulations of overexpressed protein? This concern is somewhat addressed by the EM images, but these EM images seem inconclusive – the fluorescence images have much more information. The main conclusion drawn from EM is that upon overexpression of RecA (again, not physiological conditions), the RecA goes to the membrane.

The reviewer raises an important point. As we now point out in the manuscript, RecA storage structures have been assumed to exist because of the remarkable properties of untagged RecA that enable it to aggregate in vitro. Their existence in vivo has been inferred from observations of the RecA-GFP fusion (and mutants). However, to our knowledge, there has been no demonstration that indicates that:

a) wild-type RecA forms storage structures in vivo and;

b) the RecA-GFP structures observed in vivo are in fact storage structures i.e., the RecA-GFP aggregates are not just misfolded aggregates targeted for degradation.

We agree with this reviewer that the fluorescence images reveal much more about the dynamics of RecA storage structures; we felt that the EM study demonstrated that even untagged wild-type RecA has the ability to form storage structures. Further, our study now demonstrates that the RecA-GFP can serve as an adequate marker for RecA-storage structures. We demonstrate that RecA-GFP forms storage structures in the absence of DNA damage, as well as in the presence of excess wild-type RecA.

Given that GFP fluorescence is observable even when RecA participates in storage structure formation, we propose that it is unlikely that these storage structures represent unfolded, insoluble aggregates. The observation that cells recover from UV damage in the experiments described in Figure 6G and Figure 6—video 2 support this interpretation. Further evidence that RecA can form RecA* in cells overexpressing RecA can be gathered from the observation that YPet-mCI can stain RecA bundles in the experiment described in Figure 6I and Figure 6—video 3. This finding suggests that wild-type RecA is at the very least, able to assume a RecA*-like conformation on DNA in cells expressing large amounts of RecA after UV damage.

Several studies have demonstrated that cellular copy numbers of RecA increase 2-10 fold after SOS (this is temperature/medium dependent). Our study demonstrates that RecA storage structures form under various conditions of ‘over-expression’ from 2X basal levels (as in the case of the RecA-GFP strain) to the estimated 2.5-5X SOS levels used in our experiments. The over-expression condition offers a valuable handle to study how RecA is organized in undamaged cells.

The statement that ‘… upon overexpression of RecA… RecA goes to the membrane’ is a simplistic interpretation of the properties of RecA. RecA has been demonstrated to possess an authentic ability to interact with membranes (Rajendram et al., 2015). Over-expression is not a pre-requisite for RecA association with membranes (our EM data). We argue that this is a property that is inherent to the system, and that membrane association may possibly help to stably accommodate RecA bundles (often micrometers in length) and RecA storage structures in the cell.

4) The section on the co-localization of the replication forks and the RecA structures is difficult to understand. The authors grow the cells in rich media (EZ-glycerol?) and so there are several round of replication going on at once. There are several replication forks and several RecA structures in a small cell. What is the probability that the overlap is random? Can they randomize the positions of the cells and foci and show that the overlap is not random? Alternatively, can they slow down growth and reduce number of forks and complexity and then look for overlap?

We have re-written this section to make it clearer. We have also provided chance co-localization measurements demonstrating that co-localization measurements in these experiments are meaningful in Figure 4—figure supplement 1. The Materials and methods section has been updated to include a description of the calculations.